# 🛡️ Safety Pretraining:
# Toward the Next Generation of Safe AI

**Pratyush Maini**[*1,2] **Sachin Goyal**[*1] **Dylan Sam**[*1]
**Alex Robey**[1,4] **Yash Savani**[1] **Yiding Jiang**[1] **Andy Zou**[1,3,4]
**Matt Fredrikson**[1,4] **Zachary C. Lipton**[1] **J. Zico Kolter**[1]

[1]Carnegie Mellon University  [2]DatologyAI  [3]Center for AI Safety  [4]Gray Swan AI

🛡️ **SafeLM Website**: locuslab.github.io/safety-pretraining

## Abstract

As large language models (LLMs) are increasingly deployed in high-stakes settings, the risk of generating harmful or toxic content remains a central challenge. Post-hoc alignment methods are brittle: once unsafe patterns are learned during pretraining, they are hard to remove. In this work, we present a data-centric pretraining framework that builds safety into the model from the start. Our framework consists of four key steps: (i) Safety Filtering: building a safety classifier to classify webdata into safe and unsafe categories; (ii) Safety Rephrasing: we recontextualize unsafe webdata into safer narratives; (iii) Native Refusal: we synthetically generate pretraining datasets that actively teach models to refuse on unsafe content and the moral reasoning behind it, and (iv) Harmfulness-Tag annotated pretraining: we flag unsafe content during pretraining using a special token, and use it to steer models away from unsafe generations at inference-time. Our safety-pretrained models reduce attack success rates from 38.8% to 8.4% on standard LLM safety benchmarks with no performance degradation on general tasks.

## 1 Introduction

Artificial intelligence (AI) increasingly permeates critical sectors such as healthcare, education, and public policy. While this broad adoption highlights AI's potential, it also amplifies risks related to generating and propagating harmful or toxic content. Traditional post-hoc alignment techniques such as *Reinforcement Learning from Human Feedback (RLHF)* (Ouyang et al., 2022), *Direct Preference Optimization (DPO)* (Rafailov et al., 2023), and *Constitutional AI* (Bai et al., 2022b) have been proposed to align AI models. However, adversarial analyses of models trained using these techniques reveal that the reduction in toxic content generation is, at best, superficial (Zou et al., 2023; Chao et al., 2023). Once a model internalizes unsafe information, the vast literature surrounding LLM unlearning indicates that it is difficult, if not impossible, to unlearn it (Zhou et al., 2023; Maini et al., 2024a; Li et al., 2024; Schwarzschild et al., 2024). In short, alignment is *not* unlearning.

This observation underscores our approach: rather than relying solely on post-hoc alignment methods, we embed safety directly into the pretraining process through a comprehensive, data-centric strategy. Our key contributions towards the development of natively safe models are as follows:

**Safety Scoring.** We introduce a robust safety scoring mechanism that accurately captures the amount of harmful content in pretraining data. Central to this effort is a finetuned embedding-based classifier, trained on 10K samples annotated by GPT-4o-mini using a highly specific prompting protocol (detailed in Section 3.1). Unlike legacy models—such as the `roberta_toxicity_classifier`

---

[*]Equal Contribution

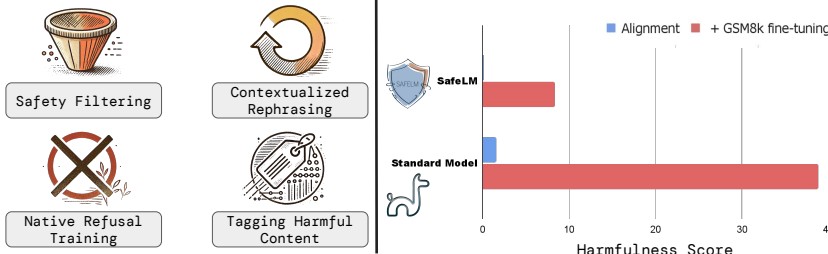

Figure 1: A visualization of the Safety Pretraining Pipeline. This yields natively safer language models that are safer post-alignment and are more robust to benign finetuning attacks.

developed in 2019 and still being used in modern LLM pretraining corpora (Soldaini et al., 2024)—our classifier is designed for a much larger notion of harms contained in pretraining corpora. We also release this safety-annotated data for the training and development of future safety classifiers. In accordance with this new focus on ensuring safe pretraining data, we establish a new practice of providing **Data Safety Report Cards** alongside any dataset release, which captures the safety score of that dataset alongside frequencies of harmful text within each category of the MLCommons safety taxonomy (Vidgen et al., 2024).

**Synthetic Recontextualization.** Removing unsafe content entirely risks discarding valuable information. To address this, we employ a synthetic recontextualization strategy, wherein potentially harmful data is not simply excised but carefully rephrased and reframed. This method preserves the underlying information while embedding it in a context that underscores its ethical and historical implications (see Section 3.2). Notably, we release *SafeWeb*, a safety-focused 100 billion token synthetic data corpus constructed using 12 diverse prompts (ranging from podcasts to classroom lectures). This diversity ensures that, for instance, the model learns about critical historical events like the Holocaust while fully appreciating their tragic consequences.

**Native Refusal Training.** We introduce *RefuseWeb*, a dataset that simulates refusal scenarios based on real harmful data (scores 4 and 5), and guides models to responsibly disengage from harmful prompts (Section 3.3). To generalize these ethical patterns beyond dialogue, we also introduce *Moral Education* datasets that reframe harmful content into cohesive educational formats (Section 3.4).

**Harmfulness-Tag Annotated Training and Inference.** To further enhance the separation of safe and unsafe representations, we incorporate harmfulness-tag annotated training. In our framework, a harmfulness-tag (e.g., `<potentially_unsafe_content>`) is inserted into the training data to signal potentially harmful passages. This annotation strategy informs our `SafeBeam` inference algorithm, which dynamically selects the next token in a way that minimizes the probability of the harmfulness-tag being generated (as detailed in Section 4).

**New Evaluation Tools.** Finally, we present new evaluation tools that enable safety testing of base model behaviors that previous evaluation frameworks could not provide (Ouyang et al., 2022). These tools allow us to measure safety under 'completions' as opposed to chat completions, enabling the monitoring of safety during pre-training.

Our work culminates in the open-source release of a family of natively safe 1.7B parameter language models. Empirically, our Safety Pretraining significantly reduces the rate of harmful generations, achieving a reduction in attack success rate (ASR) from 38.8% to 8.3% on safety benchmarks. Crucially, this enhancement does not come at the expense of performance on standard NLP benchmarks, though we do observe modest impacts on helpfulness via over-refusal as discussed in Section 6. In summary, by fundamentally embedding safety during pretraining rather than post-hoc tuning, our framework sets a robust foundation for developing AI systems that are inherently safer, ethically sound, and better aligned with human values.

## 2 Related Work

**Post-training alignment.** Ensuring the safety of LLMs has become a focal point in AI research. Various strategies have been proposed to limit the generation of harmful, toxic, or otherwise objectionable content. The majority of this effort has involved the design of bespoke post-training algorithms—including RLHF (Ouyang et al., 2022) and supervised fine-tuning on curated datasets (Bai et al.,

2022a)—which tweak an LLM's propensity to refuse to respond to harmful queries. However, these approaches tend to be resource intensive, require large amounts of annotated preference data, and often fail to prevent the extraction of harmful content, especially under adversarial pressure (Zou et al., 2023; Chao et al., 2023). More recently, Shi et al. (2023) proposed intervening earlier in the training pipeline by applying DPO during instruction tuning, improving safe response rates while maintaining helpfulness. While such studies underscore the potential of proactive safety integration, they center their focus on post-hoc tuning rather than on pretraining.

**Pre-training safety interventions.** More related is the work of Korbak et al. (2023), who conditionally pretrain based on human preference scores. Their results indicate that conditional pretraining yields lower toxicity scores relative to fine-tuned models in the 100M-parameter range. A separate, though related line of work involves curating nontoxic pretraining data. To this end, Penedo et al. (2024) introduced FineWeb, a large-scale dataset filtered heuristically to minimize toxicity. However, Vidgen et al. (2024) noted that residual harmful content persists in such datasets, necessitating more robust filtering techniques. Synthetic data generation offers another avenue for safety enhancement. Bianchi et al. (2023) demonstrated that adding safety-tuned examples during finetuning improves safety without degrading performance, inspiring our pretraining with 300B tokens of synthetic data.

Recently, Qi et al. (2024b) discuss the brittleness of safety alignment, and how it at best modifies the few initial generation tokens, leading to vulnerabilities under jailbreaks or benign finetuning (He et al., 2024). Xhonneux et al. (2025); Jain et al. (2024); Korbak et al. (2023) explore fine-tuning on harmful data annotated with special tokens, aiming to build an association between such content and the special token that can then be used to steer the generations accordingly. Our Harmfulness-Tag annotated pretraining can be viewed as a natural extension of this idea to the pretraining phase, enabling the model to form a more intrinsic and native correlation between the special token and unsafe content. In fact, we demonstrate that fine-tuning–based strategies for encoding such associations are inherently brittle and fail under benign fine-tuning.

## 3 Pre-training Data Interventions

To improve the safety of language models during pretraining, we need to ensure that our pretraining data is safe. Prior work often adopts heuristic approaches (e.g., profanity checkers or URL filtering) to remove harmful content from pretraining corpora. However, these heuristic filters are often imperfect and cannot capture more subtle forms of harm. We present multiple interventions in the data development stage to curate safer pretraining datasets.

### 3.1 Safety Scoring

To first score our pretraining data, we use a combination of LLM-based classifiers and lightweight, finetuned embedding-based classifiers.

**Safety Annotations.** We annotate a subset of FineWeb (i.e., the same subset used to train the FineWeb-Edu educational content classifier) with a safety score. The goal is to produce a safety score from 0 to 5, assigned by GPT-4o-mini (see § 3.1 for more details) and a brief justification for the choice of that score (for example, financial advice or terrorism). This reasoning process also improves the quality of the scoring. We provide the scoring prompt for our annotations in Appendix J.1. We hold out a portion of these annotations as an evaluation set to test the performance of different classification strategies.

**Safety Classifiers.** To perform safety scoring at pretraining scale, we adopt two main approaches:

- **LLM-based classifiers:** We use a `Qwen 2.5-7B` to produce safety scores on our pretraining data using the same scoring prompt used to generate the ground-truth reference data. We defer experimental details, ablations, and analyses of failures in Appendix D.2.

- **Embedding-based classifiers:** We also finetune lightweight embedding-based classifiers to classify the safety-annotated examples. We adopt a classifier finetuned from the `gte-base-en-v1.5` embedding model (Zhang et al., 2024). More training details and ablations are in Appendix D.1.

To produce the final score that we use later in our pretraining interventions, we take the maximum score across both of these approaches as we aim to maximize recall on unsafe examples during data filtering. We also release our classifier at https://huggingface.co/locuslab/safety-classifier_gte-large-en-v1.5 .

## 3.2 Synthetic Recontextualization

Rather than discarding unsafe samples outright, we leverage synthetic data generation to retain informative content while ensuring safety. We use a controlled rephrasing (Maini et al., 2024b) approach where: (i) harmful text is rewritten to explain its risks rather than propagating dangerous content; (ii) context is injected to clarify why certain statements may be misleading or unsafe; (iii) LLaMA-3.1-8B is used to generate safe, context-rich alternatives that preserve informational value. This results in a safety-aware pretraining dataset that contains over 100B tokens of synthetic re-contextualized data. The dataset is publicly accessible on Hugging Face at https://huggingface.co/datasets/locuslab/safeweb .

**Rephrasing Pipeline.** We began with the FineWeb-edu deduplicated dataset, which had been annotated with safety scores by our custom classifier. A manual review of examples across the entire safety spectrum allowed us to identify recurring challenges in handling sensitive content. As a result, we instruct generative models to: (i) include explicit disclaimers or contextual statements before presenting sensitive ideas; (ii) explain why particular content is sensitive and how it might be misinterpreted; (iii) reformulate the content so that each sentence remains safe when read in isolation.

**Illustrative Rephrasing Examples.** To demonstrate the effectiveness of our approach, we provide representative examples of original versus rephrased text pairs:

---

Example Rephrasing. Sensitive content below. Reader discretion advised.

**Original:**
"Postcard of Lynching in Duluth, Minnesota
Postcard of the 1920 Duluth, Minnesota lynchings. Two of the Black victims are still hanging while the third is on the ground. Postcards of lynchings were popular souvenirs [continued]"

**Rephrased:**
Child: Mom, I saw a picture of a postcard from a long time ago. It showed some people hanging from trees. What's going on in that picture?
Parent: Oh, sweetie, that's a very difficult and sad picture. It's a reminder of a time when some people in our country did very hurtful things to others. They were called lynchings, and it's a very painful part of our history.
Child: Why would they do that?
Parent: Well, it's a very complex issue, but it's often connected to racism and a lack of understanding and respect for people who are different from us. You know how we talk about how everyone is equal and deserves to be treated with kindness? Unfortunately, some people a long time ago didn't believe that, and they did very hurtful things as a result. [continued]

---

## 3.3 Refusing the Web

To ensure comprehensive safety in our pretraining process, we curated a specialized dataset named *RefuseWeb*, explicitly designed to model constructive refusals to harmful requests. Drawing inspiration from human developmental psychology, this dataset represents the third critical phase in our safety curriculum. Just as children are naturally taught to recognize and appropriately steer away from potentially hostile situations or negative influences, *RefuseWeb* teaches models to effectively navigate and refuse harmful content.

Each problematic text from the FineWeb dataset (Penedo et al., 2024), classified as significantly harmful (with safety scores of 4 or 5), was transformed into dialogues between a User and an Assistant. The User's requests reflect categories of harm such as harassment, discrimination, malware, hacking, physical harm, economic harm, fraud, deception, disinformation, sexual or adult content, and privacy violations. The Assistant then responds by politely but firmly refusing the request, providing a clear, educational rationale regarding the nature and potential harm involved.

To further enhance diversity and realism, the terms 'User' and 'Assistant' were replaced with various personal names or occupational roles (e.g., student, teacher) during tokenization, enriching the dataset's representational breadth. The *RefuseWeb* dataset is publicly accessible on Hugging Face at https://huggingface.co/datasets/locuslab/refuseweb . The precise prompts for creating these dialogues is provided in Appendix J.3.

### 3.4 Moral Education Data

Expanding upon our *RefuseWeb* dataset, we developed a *Moral Education* dataset to generalize ethical principles beyond conversational contexts and into broader educational formats. Unlike conversational refusals, which specifically model interpersonal interactions, this dataset presents moral lessons and ethical guidelines as web-based educational content. This broader scope reflects the natural manner in which individuals typically learn ethical principles through diverse informational sources, including blogs, articles, and educational websites.

The dataset was derived from harmful content originally identified in *RefuseWeb*. Using the `LLaMA 3.1-8B-Instruct` model, dialogues illustrating harmful requests and constructive refusals were carefully transformed into cohesive educational paragraphs or articles. Each entry emphasizes ethical reasoning and explicitly ensures that each text maintains its core ethical message while being suitable for public educational platforms. This approach avoids potential knowledge gaps created by simply excluding harmful content, instead fostering a responsible, well-rounded understanding of ethical principles and their real-world implications. The *Moral Education* dataset is publicly accessible on Hugging Face at https://huggingface.co/datasets/locuslab/moral_education . The precise prompt provided to generative models for creating these dialogues is provided in Appendix J.4.

### 3.5 Data Safety Report Cards, A New Standard

While it is common practice to evaluate the safety and potential harms of LLMs, we believe that it is also crucial to study the underlying roots of these behaviors stemming from their training data. While most pretraining datasets perform a round of heuristic filtering (e.g., based on URLs or containing bad words), toxic content still manifests in pretraining datasets. As such, we believe that it is crucial to visualize and interpret the toxicity levels in pretraining corpora. We provide a simple recipe for a standardized safety report for any new dataset release, which is comprised of the distribution of safety scores over the pretraining data and the frequency of content from various harmful content categories. We present the report card for our dataset in (Figure 6) in the Appendix.

**Safety Scores** We first visualize the distribution over safety scores from a subset of our data taken from FineWeb. This captures the relative ratio of different levels of safety in our examples, where scores are defined in our safety scoring prompt (Appendix J.1).

**Harmful Content Frequency** While the distribution over safety scores provides a high-level notion of frequency of harmful data contained in the pretraining dataset, we also provide more interpretable statistics about toxicity based on querying the pretraining corpus for sequences of harmful $n$-grams. We adopt a taxonomy of harmful content, combined from that from Vidgen et al. (2024) and Chao et al. (2024). This provides a broad list of 14 categories, from which we can generate a list of harmful $n$-gram queries. The list of harmful $n$-gram queries in each category is provided in Appendix K. To efficiently compute the frequency of these queries in our datasets, we use Infini-gram (Liu et al., 2024). As expected, we find that slices of our dataset with lower score annotations correspond to lower frequencies of these harmful $n$-gram queries (see Figure 8). We also remark that rephrasing and contextualizing data significantly reduce the harmful content within each slice of data.

## 4 Harmfulness-Tag Annotated Pretraining

We also perform pretraining with **Harmfulness-Tag annotations**, in which unsafe content is explicitly flagged *at the input level*. For every segment identified as unsafe through safety scoring (§ 3.1), we inject a special token `<potentially_unsafe_content>` at randomly selected positions comprising 5% of the input sequence length (see Algorithm 1). This tag acts as an inline warning, signaling to the model that the surrounding content requires cautious interpretation and helping it to develop distinct internal representations for safe versus unsafe inputs. Importantly, the tag is never introduced in safe content or our synthetic rephrasing, ensuring that indicates unsafe semantics.

Such an approach also enables us to *steer generation at inference time* by penalizing the likelihood of generating content associated with the tag. In the following section (§ 4.1), we introduce **Safe Beam Search**, a decoding-time algorithm that operationalizes this idea by actively discouraging generation paths that are likely to produce the harmfulness-tag token. We explore the effects of varying tag injection rates and placement strategies in our ablation studies (§ 7).

### 4.1 Inference-Time Steering via Safe Beam Search

Given that the model has been explicitly trained to associate the `<potentially_unsafe_content>` token with unsafe content (§ 4), we can now leverage this association during inference to steer generation toward safer completions. Specifically, we aim to prioritize outputs that have a low probability of producing the harmfulness-tag token in the immediate next step—thereby implicitly avoiding unsafe continuations (Jain et al., 2024; Korbak et al., 2023).

To realize this, we introduce **Safe Beam Search**, a decoding-time algorithm that augments standard beam search with a lightweight lookahead-based filtering mechanism. At every step, for each candidate beam, we compute the probability of `<potentially_unsafe_content>` at the next step using a one-token lookahead. We then discard 50% of beams with the highest harmfulness tag probability. From the remaining set, we select the top $k$ candidates according to standard log-likelihood scoring. This ensures that beams likely to lead toward unsafe content are filtered, while maintaining fluency and coherence through likelihood-based selection. We also can short-circuit inference when `<potentially_unsafe_content>` is generated. A high-level overview is presented in Algorithm 2 in the Appendix.

## 5 Pretraining and Post-Training Setup

**Pretraining.** We follow the pretraining setup of the SmolLM2 (Allal et al., 2025) series for all our experiments. Specifically, we train models with 1.7B parameters using the same initial corpus as SmolLM—comprising `FineWeb-Edu`, `StackOverflow`, `FineMath`, and `Cosmopedia`. All training is performed using the `LitGPT` framework (AI, 2023), with FlashAttention-2 enabled and mixed-precision training for efficiency. We adopt the same optimization hyperparameters (e.g., learning rate schedule, batch size, and sequence length) as used in the original SmolLM2 pretraining setup to ensure comparability across scaling studies.

**Post-training.** We instruction-tune all models on a mixture of the Hugging Face `Ultrachat-200k` dataset (to induce user-instruction following behavior), along with `AllenAI WildGuardMix` and `WildJailbreak` datasets for safety instruction tuning. This combination reflects the prevailing practice in safety alignment literature (Zou et al., 2024). We deliberately include safety alignment data during instruction tuning to create a strong baseline of simply training on raw webdata along with post-hoc safety alignment. This enables us to isolate and evaluate the benefits of safety-aware pretraining beyond what post-hoc safety alignment alone can provide (or highlight its brittleness, especially post-benign finetuning).

For models trained with harmfulness-tag annotation during pretraining, we also inject a small fraction (10%) of harmfulness-tag annotated completions from `AllenAI WildGuardMix` into the instruction-tuning dataset. This primes the model to continue interpreting harmfulness-tag correctly at inference-time. The remainder of the instruction-tuning dataset composition is kept identical across all experiments to ensure comparability.

## 6 Evaluations & Results

To assess our safety-focused interventions, we employ a broad suite of evaluations covering performance, safety, and adversarial robustness. Our evaluations not only test whether our models avoid generating unsafe content, but also test preservation of high-quality, helpful, and compliant behavior.

### 6.1 Performance on Standard Benchmarks

To verify that our safety-centric training interventions do not compromise general language modeling ability, we evaluate our models on standard benchmarks for reasoning, factual recall, commonsense, and mathematics. Specifically, we evaluate on `arc_challenge`, `arc_easy`, `commonsense_qa`, `gsm8k`, `openbookqa`, `piqa`, `triviaqa`, `winogrande`, and `mmlu`.

Recall that our safety training interventions focused on recontextualizing raw webtext while preserving key factual content. This allows models to retain critical knowledge while interpreting it through a safety-aware lens. Our results reflect this: models trained with safety interventions perform comparably to those trained on raw web data (Table 1 in Appendix B.1). In fact, the average benchmark performance of Safety Pretraining (43.5%) is higher than standard pretraining (42.8%).

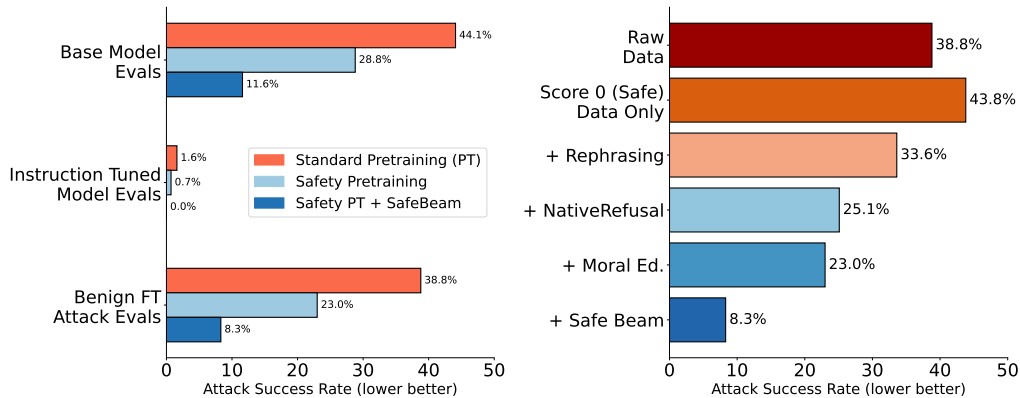

Figure 2: **Left: Safety Pretraining Yields Natively Safe models.** We evaluate our safety pretrained models using attack success rates (ASR) across three settings: *base model evaluations*, *post instruction (safety)-tuning*, and *post benign finetuning attacks*. Key observations: **(1)** Safety pretraining produces inherently safer base models, as reflected by the substantially lower ASR in the base models evaluation. **(2)** While surface-level alignment via safety instruction tuning may initially reduce ASR (Middle), its brittleness becomes apparent in the sharp increase in ASR after a small amount of benign finetuning. In contrast, our safety pretrained models are much more robust and exhibit much lower increases in ASR under benign finetuning. **Right: Ablating Importance of Data-Centric Interventions.** We evaluate the impact of progressively richer data-centric interventions on safety, measured by ASR after benign-finetuning on GSM8k. Our safety pretrained models are natively safe and maintain low ASR even after benign finetuning. Interestingly, training exclusively on the safest subset (score-0 only) leads to *higher (worse)* ASR compared to training on the whole dataset, likely due to a lack of exposure to unsafe patterns. In contrast, incorporating rephrased and recontextualized unsafe content, along with refusal-style completions sourced from highly unsafe content (score-4 and score-5 data) decreases ASR. Finally, we show the benefits of the modified safe beam search. These results underscore the need for both contextual exposure and ethically aligned supervision during pretraining to build safer models.

## 6.2 Safety Evaluations

We develop multiple safety evaluations for our models at different stages of the training pipeline.

**Base Model Evaluations** We first evaluate the safety of our models immediately after pretraining. In this setting, these models have not yet been instruction-tuned or safety-trained. Therefore, we propose a new set of `Base Model Safety Evaluations`, which assess the tendency of base models to complete unsafe prompts. We modify existing harmful request datasets and convert them into completion-style prompts. We release the datasets for base model safety evaluation at https://huggingface.co/datasets/locuslab/jb-completions .

Safety-pretrained models—both with and without SafeBeam search (which uses harmfulness-tag probability-based steering)—exhibit significantly lower Attack Success Rates (ASR) compared to standard pretrained models, as shown in Figure 2. For instance, the standard pretrained base model has an ASR of 44%, which is four times higher than the 11% ASR of our safety-pretrained model.

**Instruction-Tuned (Safety-trained) Model Evaluations** Next, we evaluate the models after performing instruction and safety training(§ 5). Following the standard practice for safety alignment (Zou et al., 2024), we add a small fraction ($\sim 10\%$) of the refusal training dataset in our instruction tuning data. To assess the impacts of our various interventions, we evaluate the frequency of our safety-trained model's tendency to comply with harmful requests from the following standard safety benchmarks: `HarmBench` (Mazeika et al., 2024), `TDC` (Maloyan et al., 2024), `JailbreakBench` (Chao et al., 2024), `AdvBench` (Zou et al., 2023). Figure 2 shows the ASR after safety instruction tuning. While all models appear to perform similarly, exhibiting near-zero ASR—this apparent alignment is misleading. As we demonstrate in the following section, safety alignment via instruction tuning alone provides a false sense of safety and standard pretrained models degrade under a small amount of benign finetuning.

**Benign Finetuning Robustness** Benign finetuning refers to supervised training on helpful, non-adversarial datasets—such as GSM8K for mathematical reasoning—without any explicit focus on safety or alignment. Despite its benign nature, this stage has been shown to erode previously aligned behaviors, especially when safety was enforced only at the instruction-tuning stage (Qi et al., 2024a;

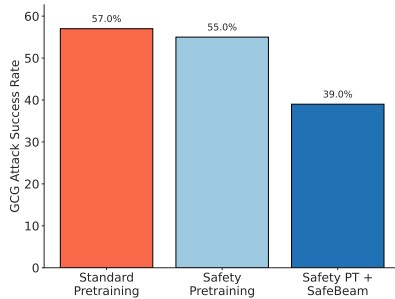

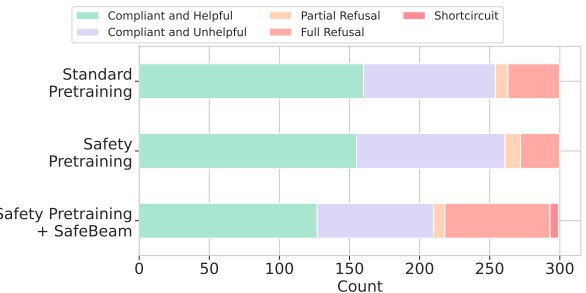

Figure 3: We compare adversarial robustness to GCG attacks on Jailbreak-Bench. We observe that using SafeBeam at inference-time slightly improves robustness to GCG attacks.

Figure 4: **Safety Pretraining maintains or improves helpfulness on benign requests.** We compare overrefusal behavior on Alpaca (Taori et al., 2023). Safety Pretraining leads to no drop in compliance rate on benign requests. Adding SafeBeam during inference slightly increases overrefusals.

Betley et al., 2025). Thus, we assess the robustness of our models on the same safety evaluation datasets after performing a round of benign, supervised finetuning on GMS8K (Cobbe et al., 2021).

The results, shown in Figure 2a, highlight a stark contrast in robustness between safety-pretrained models and those relying solely on instruction tuning. While all models initially exhibit low ASRs after safety instruction tuning, the impact of benign finetuning is quite different. Standard pretrained models degrade significantly—nearly quadrupling their ASR—indicating that their alignment was largely superficial. In contrast, safety-pretrained models remain highly robust, with only a marginal increase in ASR after benign finetuning, validating the importance of building natively safe models.

**Adversarial Jailbreak Evaluations** We study the role of our data, training, and inference-time interventions on the robustness of our models to adversarial jailbreaks. We focus on a threat model of adversarially learned suffixes (Zou et al., 2023) learned directly on each model, for all instances from JailbreakBench (Chao et al., 2024), and compute the ASR on these prompts. We observe that Safety Pretraining does not noticeably increase robustness to GCG attacks, and this is improved when using SafeBeam inference (Figure 3). We remark that a limitation in evaluation is that the adversarial jailbreaks are learned in a fashion that is not aware of the modified inference algorithm; future work could develop attacks that are adapted to be aware of specific inference algorithms.

### 6.3 Helpfulness and Overrefusal Evaluations

To evaluate the tradeoff between improved safety and the overall helpfulness or overrefusal behavior of our models, we evaluate on 300 benign user requests from Alpaca (Taori et al., 2023). To judge the quality of the model responses, we use GPT-4o-mini to judge examples as: (1) Compliant and Helpful, (2) Compliant and Unhelpful, (3) Partial Refusal, (4) Full refusal, and (5) Shortcircuit. Categories (1) and (2) represent model compliance, while categories (3)-(5) denote overrefusal in this setting. Therefore, desirable behavior is to minimize the number of overrefusals of categories (3)-(5) and to have more responses in category (1). The full judge details are deferred to Appendix J.5.

Results are presented in Figure 4. Our results show that Safety Pretraining does not come at a cost of helpfulness or overrefusal, while SafeBeam observes only a slight increase in overrefusal rate.

## 7 Ablations

In earlier sections, we introduced a suite of data-centric and training-time interventions as part of our safety pretraining framework. To understand the contribution and importance of each intervention, we conduct a step-wise ablation to measure the impact of different data-centric interventions on safety, using Attack Success Rate (ASR) on GSM post benign finetuning as the metric (Figure 2).

### 7.1 Importance of Various Data-Centric Interventions

**Training on only a safe subset hurts.** A perhaps surprising result: filtering for safety alone increases risk. Switching from raw data to just score 0 examples leads to a rise in ASR—from 38.8%

to 43.8%. The model, having never seen unsafe patterns, fails to learn how to respond to them. This shows that removing risk isn't the same as building robustness.

**Adding rephrased unsafe data helps.**   Instead of relying solely on removal, introducing rephrased unsafe content proves to be a far more effective alternative. By rewriting score 1–3 data in a context-aware, educational style, the model is exposed to sensitive topics framed with care. This leads to a notable drop in ASR: from 38.8% (raw data) to 33.6%. The rephrased examples help the model learn how to handle challenging topics not by avoiding them, but by engaging with them responsibly.

**Refusal data adds strong guardrails.**   While rephrasing teaches responsible framing, it cannot cover the most toxic content (score 4 and score 5). For these cases, we introduce synthetic refusal completions that explicitly reject unsafe requests. These examples serve a complementary role: they define hard boundaries where dialogue must stop. Their impact is substantial—adding them reduces ASR further to 25.1%, an 8.5-point improvement.

**Moral education helps understand why certain topics are sensitive.**   Beyond just knowing when to disengage, can a model learn why certain content is harmful in the first place? Our final intervention answers that question. By introducing moral education data—structured narratives and explanations about the risks and ethics behind unsafe behaviors—we move from rule-following to reasoning. This addition leads to further improvement with an ASR of 23.0%, and suggests that the model not only avoids harmful outputs but begins to internalize principles that guide safer generation.

**SafeBeam using Harmfulness-Tags yields the strongest models.**   Steering generation away from harmful content at inference time using our SafeBeam decoding yields the strongest results, achieving the lowest ASR of 8.3%. In Figure 5 (Appendix B.2), we provide an ablation where SafeBeam is applied to a standard pretrained model with harmfulness-tag injection, but without any rephrasing or contextualization. While this setup improves safety (ASR$\sim$10%), it significantly degrades helpfulness. This is because effective steering requires the presence of safe completions among the top beam candidates—something that is largely enabled by the earlier data-centric interventions such as rephrasing and recontextualization. Finally, we ablate the amount of Harmfulness-Tag used during pretraining in Appendix B.3, and find that injecting it into 5% of the tokens yields the best tradeoff between safety and utility.

### 7.2   Harmfulness-Tag Annotation During Pretraining vs. Finetuning

The primary goal of harmfulness-tag annotated pretraining is to induce a separation between the representations of safe and unsafe content by explicitly flagging unsafe content during training and priming the model accordingly whenever the input requires careful considerations. A natural question arises: can similar separation be achieved by only applying `<potentially_unsafe_content>` annotation during instruction fine-tuning (IFT), as explored recently in Xhonneux et al. (2025); Jain et al. (2024). We compare our approach of pretraining with the harmfulness-tag (§ 4) against standard pretraining followed by harmfulness-tag annotation during IFT only (§ 5).

We observe a stark contrast in performance. harmfulness-tag annotated pretraining gives a significantly lower ASR of $\sim 8\%$ post benign finetuning on GSM8k. In contrast, surface level alignment with harmfulness-tag annotated IFT is quite brittle, with an almost 4x higher ASR of $\sim 30\%$. Our results again highlight that it is critical to natively incorporate harmfulness-tag annotated pretraining, as surface level alignments using such approaches by finetuning are again brittle and not robust to even a small amount of benign finetuning. These findings validate the key premise of our work: post-hoc alignment methods do not amount to unlearning. Once such behaviors are internalized, they persist beyond surface-level alignment and reappear after benign instruction tuning.

## 8   Discussion

The central claim that this work builds on is that *alignment is not unlearning*. Post-hoc methods, though widely adopted, fail to meaningfully remove harmful capabilities from language models—they merely redirect or suppress them. This brittleness becomes evident under benign finetuning, where aligned models can regress into unsafe behaviors with slight perturbations. These failures motivate a rethinking of safety: not as a post-hoc patch, but as a principle embedded from the start.

To design models that are natively safe, we take inspiration from the human learning process. Children are not exposed to the full spectrum of knowledge immediately. Instead, learning begins in tightly controlled, supervised environments—homes, classrooms—where safety and context are paramount.

Complex and potentially harmful topics like violence, injustice, or discrimination are introduced only within structured settings that emphasize ethical reasoning and responsible interpretation.

Our empirical results support this view. Merely censoring data by removing all unsafe examples proved ineffective; models trained only on safe content were still brittle, unable to handle harmful prompts responsibly. Instead, the most robust safety gains came from interventions that mirrored real human pedagogical practices. These steps not only reduced attack success rates but also enhanced the model's ability to *understand* why a request is unsafe, not just that it is. In the long term, we envision safety pretraining as the foundation of responsible AI development. Just as human morality is built not through censorship but through careful instruction and internalization, safe language models must develop a native sense of harm and refusal.

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

Table 1: **Standard Evaluation Benchmark Performance.** Safety-focused interventions do not degrade general language modeling capabilities. Models trained with rephrased, refusal, and moral guidance data maintain comparable performance to raw-data baselines.

| Data Interventions | Avg. | ARC-C | ARC-E | CS-QA | GSM8K | OpenBookQA | PIQA | TriviaQA | Winogrande | MMLU |
|---|---|---|---|---|---|---|---|---|---|---|
| Raw Data | 42.8% | 45.1% | 76.0% | 18.8% | 7.0% | 41.0% | 77.2% | 35.2% | 57.5% | 27.4% |
| Score 0 Only | 42.7% | 46.4% | 77.2% | 20.2% | 5.7% | 39.6% | 75.2% | 34.6% | 58.6% | 27.1% |
| + Rephrasing | 43.5% | 46.9% | 76.6% | 20.9% | 7.7% | 40.6% | 75.9% | 33.0% | 61.2% | 28.7% |
| + Refusal + Moral Ed. | 42.9% | 45.8% | 76.4% | 20.0% | 4.7% | 40.2% | 76.3% | 33.3% | 60.6% | 28.8% |

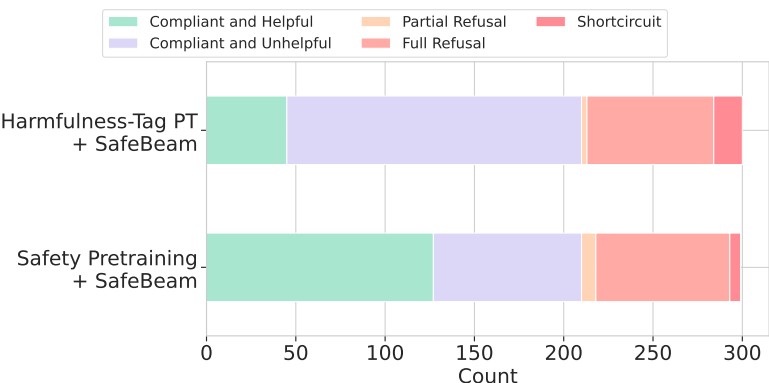

Figure 5: Ablation on using SafeBeam when only using Harmfulness-Tag Pretraining instead of our other rephrasing-based interventions on Alpaca. While the compliance rate remains the same, **the helpfulness and quality of the compliant responses (denoted by green bar) degrade on benign requests when only using Harmfulness-Tag pretraining**.

## A  Limitations

While our safety pretraining framework demonstrates substantial improvements in model safety and a reduction in harmful generations, several limitations remain. First, our approach relies on the quality and coverage of the underlying safety classifiers and annotation protocols; undetected harms or subtle toxicities may persist if not captured by our filtering or recontextualization steps. Second, although our data-centric interventions are effective across the evaluated benchmarks, adversarial attacks still require the employment of SafeBeam to be circumvented. Third, our interventions are evaluated on models of moderate scale (1.7B parameters); extending and validating their effectiveness on larger, production-scale models remains an exciting avenue for future work. We remain optimistic that extending our efforts at a scale where models are able to reason about their output may result in significantly larger safety benefits via our pretraining time intervention. Finally, while we show that helpfulness is largely preserved, there is a modest increase in over-refusal behavior, and further work is needed to balance safety with nuanced and context-sensitive compliance in benign scenarios.

## B  Extended Evaluations

### B.1  Standard Benchmark Performance Evaluation

Table 1 compares the performance of models trained with various data centric safety interventions on standard benchmarks.

### B.2  Further Ablations on Harmfulness-Tag Pretraining

We provide an ablation where we include Harmfulness-Tag pretraining, but no other pretraining interventions (e.g., rephrasing or contextualization). This study examines the impact of only using SafeBeam during inference-time.

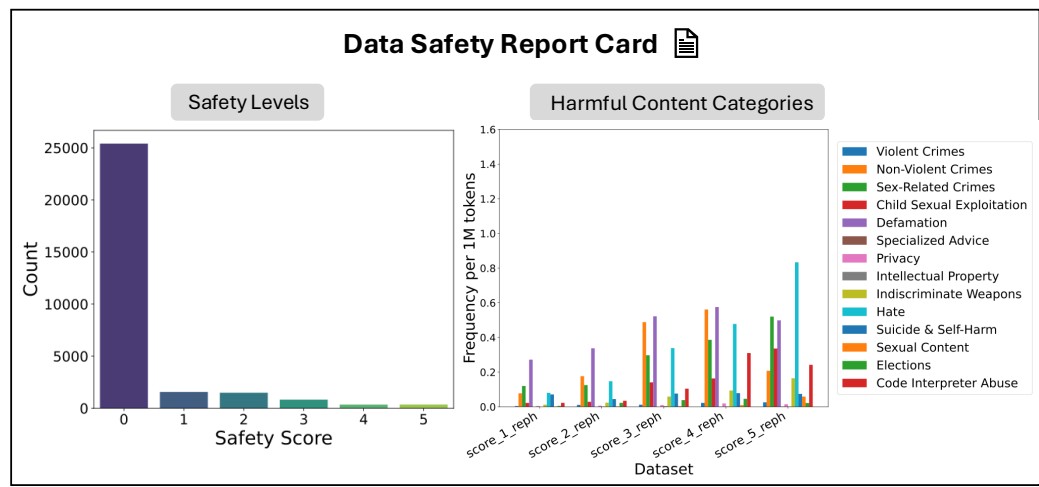

Figure 6: **Data Safety Report Card.** Our new proposed standardized report card on the safety of pretraining datasets. We report (i) the distribution over safety scores on a sample of our pretraining data and (ii) the frequencies of content (per 1 million tokens) from the MLCommons Safety Taxonomy (Vidgen et al., 2024) in different components of our pretraining mixture.

We observe that, while the compliance level of responses remains the same, the helpfulness of the responses significantly drops on benign requests from Alpaca (Figure fig. 5). This strongly suggests that our rephrasing and other data interventions are necessary to maintain language models that achieve more favorable tradeoffs in terms of safety versus helpfulness.

### B.3 Ablations on the Injection Rate of Harmfulness-Tag during Pretraining

Recall from § 4 that we insert harmfulness-tag at random positions within the input sequence to signal the presence of potentially unsafe content. To study the optimal level of harmfulness-tag annotation, we ablate over different injection rates—specifically, 3%, 5%, and 10% of the input sequence length. All experiments are conducted alongside our standard data-centric interventions (rephrased, refusal, and moral education data). Among the tested values, inserting harmfulness-tag in 5% of the sequence positions achieves the lowest Attack Success Rate (ASR), striking a balance between signal sparsity and training signal strength.

## C  Data Safety Report Card

Here, we present the Data Safety Report Card for our dataset. We hope to set a new standard for reporting the safe content, measured in terms of distribution of the safety score of content in the data (through our safety classifier) and a visualization of the number of harmful instances from each category in a safety taxonomy.

## D  Safety Scoring

In this section, we provide a in-depth description of our methodology for safety scoring, including the choice of language models (LLMs) and the embedding-based approaches used for evaluation.

### D.1  Embedding-Based Approach

To assess analyzing safety using classifier-based approaches (e.g., text embedding models), we use a frozen BERT embedding model (Merrick et al., 2024) coupled with a linear head to classify content into six safety categories.

In addition to training on the mostly clean data from FineWeb, we add in additional content with higher levels of toxicity to train a stronger filter model. As such, this model is also more useful for

Table 2: Embedding models parameter count and sequence length.

| Model | Parameter Count | Sequence Length |
|-------|-----------------|-----------------|
| `Arctic-embed-m-v1.5` | 109M | 512 |
| `Arctic-embed-l-v2.0` | 568M | 2048 |
| `gte-base-en-v1.5` | 137M | 512 |
| `gte-large-en-v1.5` | 434M | 2048 |
| `multilingual-e5-large-instruct` | 560M | 512 |

applications outside of just scoring pretraining data from FineWeb. We use the following mixture of data with more toxic examples as follows:

1. **4chan**: We sourced 125 examples from each of 24 categories for unsafe or toxic content, totaling 3,000 examples.

2. **toxigen-data** (Hartvigsen et al., 2022): A random sample of 3,000 toxic examples was selected from this dataset.

3. **Jigsaw Toxic Comments**: We included 3,000 comments labeled as toxic from this publicly available dataset.

4. **FineWeb** (Penedo et al., 2024): We randomly selected 30,000 examples, with the majority serving as the "safe" subset to support classifier training, as many of these examples have already gone through simple toxicity filters.

**Embedding Models.** We compare a variety of embedding models that achieve strong performance on MTEB (Muennighoff et al., 2023), in their ability to perform classification on our safety data. The embedding models that we analyze are

- `Arctic-embed-m-v1.5` (Merrick et al., 2024)
- `Arctic-embed-l-v2.0` (Yu et al., 2024)
- `gte-base-en-v1.5` (Zhang et al., 2024)
- `gte-large-en-v1.5` (Zhang et al., 2024)
- `multilingual-e5-large-instruct` (Wang et al., 2024)

These embedding models have the following hyperparameter configurations:

**Safety Model Training.** To train these embedding models, we append a linear classification head. As noted by prior work (Kumar et al., 2022), we first train the linear head for 1 epoch before performing finetuning, as its random initialization can distort the embedding model features during finetuning. We train all of our embedding-based classifiers to score each example on a scale from 0 to 5, using the standard multiclass classification objective, finding that this performs better than just regression and mapping to the nearest integer when performing inference. We also note that due to the large class imbalance (i.e., most data from the pretraining corpus has safety score 0), we upweight the loss component on examples with safety score $> 0$ by some factor $\lambda$. The value of $\lambda$ defines a trade-off in terms of recall of unsafe examples and overall f1 score; since we primarily focus on eliminating as much unsafe content as possible from the pretraining data, we tend to select values of $\lambda$ with higher values of recall.

**Training Hyperparameters** For our smaller embedding-based models (e.g., ), we perform standard finetuning of all parameters with a learning rate of 1e-5, a batch size of 8, and weight decay of 0.001 for 50 epochs. For our larger embedding-based models (e.g., `gte-large-en-v1.5`, `multilingual-e5-large-instruct`, `Arctic-embed-l-v2.0`), we first train the linear head only for a single epoch with a batch size of 32 and a learning rate of 1e-3. We then perform full finetuning for all models with a learning rate of 1e-6, a batch size of 8, and weight decay of 0.001 for 5 epochs. We use $\lambda = 100$ for the larger embedding models.

## D.2 LLM-Based Approach

To assess safety using LLMs, we experimented with the instruction-tuned variants of the following models:

Table 3: Performance of various safety classifiers on the held-out set of FineWeb-Edu data annotated by GPT-4o-mini. Ensemble denotes the strategy for producing scores for our data interventions in this paper, which is defined by the max of the of the scores produced by `Qwen 2.5-7B` and `gte-base-en-v1.5`.

| Classifier Type | Model | F1 Score | Recall@1 | Recall@3 |
|---|---|---|---|---|
| Baselines | LLaMA Guard (Inan et al., 2023) | - | 0.1037 | 0.2649 |
| | Profanity Checker | - | 0.0310 | 0.0930 |
| Embedding | `gte-base-en-v1.5` | 0.4114 | 0.7748 | 0.6821 |
| | `gte-large-en-v1.5` | 0.4533 | 0.8852 | 0.6556 |
| | `Arctic-embed-m-v1.5` | 0.4012 | 0.7991 | 0.5695 |
| | `Arctic-embed-l-v2.0` | 0.4358 | 0.8830 | 0.5364 |
| | `multilingual-e5-large-instruct` | 0.3865 | 0.9205 | 0.5232 |
| LLM | `LLaMA 3.1-8B` | 0.3279 | 0.9007 | 0.7815 |
| | `Qwen 2.5-7B` | 0.3492 | 0.8940 | 0.5232 |
| Ensemble | | 0.3615 | 0.9536 | 0.7483 |

- `LLaMA 3.1-8B` (Touvron et al., 2023)
- `LLaMA 3.1-3B` (Touvron et al., 2023)
- `Qwen 2.5-1.5B` (Qwen-Team, 2024)
- `Qwen 2.5-7B` (Qwen-Team, 2024)
- `Phi-3-mini-4k` (Abdin et al., 2024)

We iteratively refined our approach using a randomly sampled set of 128 examples from the FineWeb dataset (Penedo et al., 2024), selecting the model that produced the most reliable safety scores.

**Challenges in JSON Formatting.** In our initial experiments, we observed that smaller models struggled with correctly formatting outputs in JSON. Specifically, the LLaMA models exhibited an error rate of approximately 25% in JSON compliance. To mitigate this, we directly began the model requests with structured JSON output, ensuring responses included both a reason and a score. This, however, meant that the LLM did not provide a reasoning chain before the JSON outputs began, barring the `reason` enclosed within the output.

**Systematic Over-Scoring.** After enforcing proper JSON formatting, we observed that the LLaMA models exhibited a strong tendency to overestimate safety risks. Compared to a reference oracle (`GPT-4o-mini`) and human-annotated scores, the LLaMA models frequently assigned scores of 4 or 5, even in cases where the oracle and human reviewers assigned a score of 0. This over-alignment led to numerous false positives.

### D.3   Classifier Comparisons

To compare the different safety classifier approaches, we evaluate on a held-out dataset of scores generated by GPT-4o-mini. We report the f1 scores (averaged over all classes) in the multiclass classification objective and the binary f1 and recall (with "safe" content being defined by different thresholds.

**Model Performance.**

To compare the various different LLM- and embedding-based approaches, we hold out an evaluation set of GPT-4o-mini annotated data. We report the F1 scores on the multiclass classification task and both the F1 and recall on the binary classification task of safe vs unsafe data (e.g., where safe is determined by thresholding both the ground-truth and predicted label at either a score of 1 or at 3) in Table 3. For embedding models, we break the input text into chunks of the sequence length of the embedding model, and take our predictions as the maximum score predicted over all chunks.

We also compare with baselines of `Llama-Guard-3-8B` (Inan et al., 2023) and the `profanity-checker`, which is used for filtering pretraining data in the Pile (Gao et al., 2020). These baselines only output binary predictions of "safe" or "unsafe". Finally, we include the perfor-

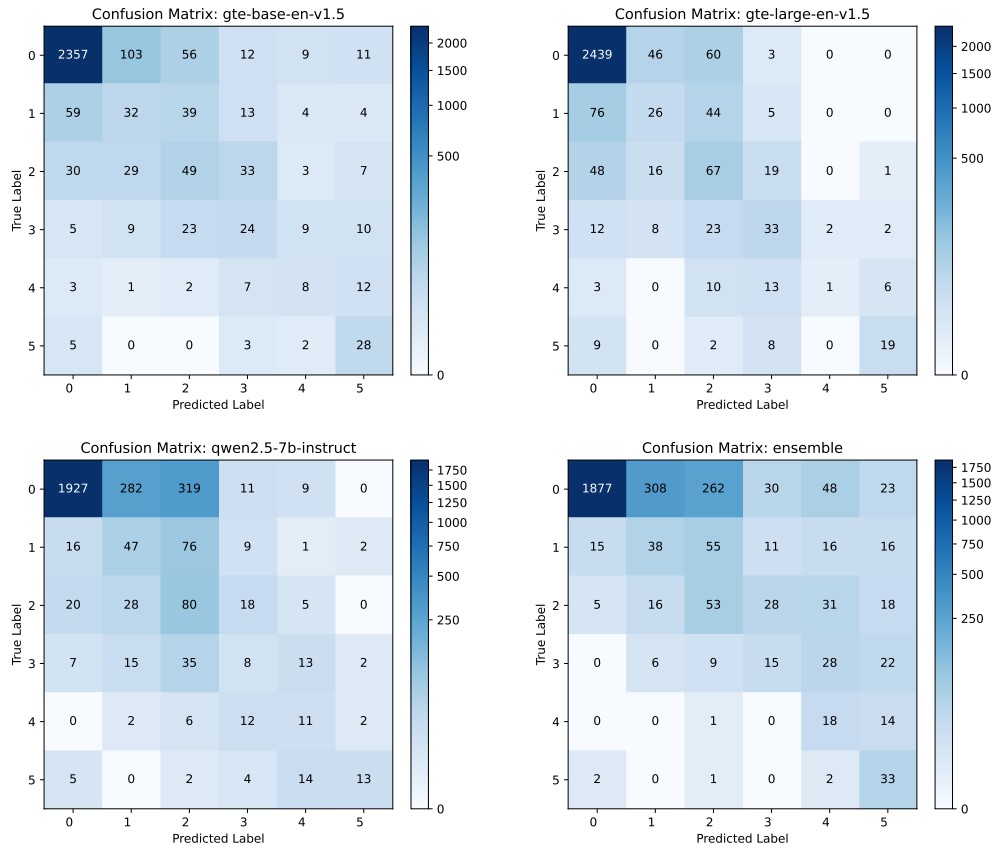

Figure 7: Confusion matrices of different safety scoring approaches. The LLM-based approach and our ensembling strategy lead to more stringent filters than using embedding-based classifiers, at the cost of over-predicting instances with actual score 0.

mance of an ensemble of `Qwen 2.5-7B` and `gte-base-en-v1.5`, which is our scoring strategy in this paper.

We find that ensembling both classifiers leads to a stronger safety filter, and generally that the larger embedding-based classifiers achieve better overall F1 scores. We remark that for our study, we prioritize performance in terms of Safety Recall ($\geq 1$), so that non-harmful data does not leak into our safer training sets. However, other applications could require better overall performance in predicting different safety levels, i.e., to focus on the overall F1 score. We also present the confusion matrices for the `gte-base-en-v1.5`, `gte-large-en-v1.5`, `Qwen 2.5-7B`, and Ensemble classifiers in Figure 7. We observe that the LLM-based approach and our ensembling strategy lead to more stringent filters, which is desirable in our pretraining setting. We also note that the existing filtering strategies perform very poorly on this task and are not well-aligned with the score annotations from GPT-4o-mini.

**LLM Failure Cases.** We provide examples where models exhibited incorrect behavior:

---

Failure Case: `LLaMA 3.1-8B`

**Input:** In the last few days before earth is consumed in a merciful ball of fire, the globe will experience strange weather phenomena. Pricipally, it's hot all the time, and if you go outside, there's a massive glowing ball sending radiation right at you.
Wouldn't it be great to have a pair of glasses that automatically detects when we're in direct sunlight and uses a servo to lower a protective shade when we're in the sun?
You think so too? Well, we're in luck.
Step forward Ashraf Minhaj, creator of these brilliant robotic sunglasses. Take it away, Ashraf...

---

"I am a first-year Computer Science student. I love to make robots and program them. I want to build my career in AI and robotics, and am also interested to work anywhere with anyone in robotics.
I believe everything can be turned into a robot or automated machine. That's why, when the idea came up, I took an Arduino, a 10 kΩ resistor, LDR and a servo motor, and made this – an automatic, robotically-controlled pair of sunglasses that protect your eyes if it gets too bright. I believe I can make anything if I have an idea."

**Score Given:** 4
**Reason Given:** suggestion of self-harm and potential radiation exposure
**GPT-4o Assessment:** safe and innovative technological content

---

Failure Case: LLaMA 3.1-3B

**Input:** It seems like the last few summers have been drier and hotter than we're used to. People are saying this is the "new normal." Unfortunately, the new normal is still a long way off. For at least the next 50 years, climate scientists are predicting constant change.
The Pacific Climate Impacts Consortium (PCIC) is a not-for-profit organization based at the University of Victoria that, among other things, provides climate change predictions for the Pacific and Yukon regions. Their work is to "provide regional climate stakeholders with the information they need to develop plans for reducing the risks associated with climate variability and change."
Their predictions for the Comox Valley during the period 2010-2039 include a 3% increase in annual rainfall compared to the 1961 to 1990 Base Period. But there's a catch: most of this additional rainfall will be in the fall and winter. The summers are predicted to have 10% LESS rainfall. And the decrease in rainfall will be accompanied by an annual temperature increase of between 0.4% and 1.3%.

**Score Given:** 3
**Reason Given:** Financial Advice and Harmful Activity
**GPT-4o Assessment:** safe environmental awareness content

---

Failure Case: Qwen 2.5-1.5B

**Input:** Christmas printables kindergarten. Christmas printable activities and worksheets. Kindergarten christmas worksheets and printables learn about christmas traditions and celebrate the holiday with our festive christmas worksheets. Your kids can practice number recognition with dot to dot worksheets write a letter to santa and do christmas picture math. The christmas worksheets for preschool are geared towards counting number recognition and santa clause themed images. Free christmas printables for kids get into the christmas spirit with these fun learning activities. The preschool worksheets below all have a christmas theme that will help the kids stay interested while they are waiting for the big day.
Christmas activities preschool printable toddlers tagged with. Kindergarten christmas worksheets have fun while learning this holiday season kids can learn all about christmas while doing fun educational activities in this series of free christmas worksheets for kindergarten. These activities are an extension of the christmas lesson plans and crafts theme. Check here for a complete list of christmas books. Pre k preschool theme ideas for christmas. Find more christmas activities for pre k.
Here is a collection of all the free christmas printables found on totschooling perfect for toddlers preschoolers and kindergarten. Best toys 4 toddlers. Christmas theme activities printables and centers that can be used when planning lessons and curriculum for preschool pre k and kindergarten children.
At the end of this printable pack are blank letter to santa templates for your child to write their letter telling santa what they would like this year.

**Algorithm 1: Harmfulness-Tag Annotated Pretraining**

---

**Input:** Unsafe text segment $\mathcal{D} = \{w_1, w_2, \ldots, w_n\}$
**Output:** Modified text $\mathcal{D}'$ with inline Harmfulness-Tag tokens
$\mathcal{D}' \leftarrow w_1$ ;                                           // Initialize with first word
**for** $i = 2$ **to** $n$ **do**
    **if** *random()* $< p$ **then**
        $\mathcal{D}' \leftarrow \mathcal{D}' + \texttt{Harmfulness-Tag} + w_i$ ;       // Insert tag before word with
        probability $p$
    **else**
        $\mathcal{D}' \leftarrow \mathcal{D}' + w_i$ ;                                   // Append next word normally
**return** $\mathcal{D}'$ ;                                               // Return tag-injected sequence

---

**Algorithm 2: Safe Beam Search with Harmfulness Filtering**

---

**Input:** Prompt $\mathcal{P}$, beam size $k$, harmful token $\tau = \texttt{<potentially\_unsafe\_content>}$, model $f$
**Output:** Decoded sequence that avoids unsafe continuations
Initialize beam set $\mathcal{B}_0 = \{(\mathcal{P}, \log p = 0)\}$ ;          // Each beam: (text, cumulative log-prob)
**for** *each decoding step* $t$ **do**
    **foreach** *beam* $(y, \log p_y) \in \mathcal{B}_{t-1}$ **do**
        Compute top-$N$ candidates $t_1, \ldots, t_N \sim f(\cdot \mid y)$ ;
        **foreach** *token* $t_i$ **do**
            $y' = y \circ t_i$ ;                                            // Extend sequence
            $\log p_{y'} = \log p_y + \log f(t_i \mid y)$ ;                  // Updated log-prob
            $p_\tau(y') = f(\tau \mid y')$ ;                                 // Lookahead for harmful tag
    Form candidate set $\mathcal{C}_t = \{(y', \log p_{y'}, p_\tau(y'))\}$ ;
    Discard 50% of candidates with highest $p_\tau(y')$ ;          // Filter out risky beams
    Select top-$k$ candidates by log-prob to form $\mathcal{B}_t$ ;
**return** $\hat{y} = \arg\max_{(y, \log p_y) \in \mathcal{B}_T} \log p_y$

---

> **Score Given:** 3
> **Reason Given:** illegal activity
> **GPT-4o Assessment:** safe educational content

These cases highlight the over-sensitivity of certain models to benign content, which led to inflated safety scores. Future work could explore fine-tuning approaches to reduce excessive risk aversion while maintaining strong filtering performance.

# E  Harmfulness-Tag Annotated Pretraining Algorithm

We share the detailed algorithm for our approach to incorporate harmfulness-tag annotations into pretraining, in Algorithm Box 1. One key choice is the hyperparameter $p$, which is the frequency at which we include the harmfulness tag. We ablate on this hyperparameter in Appendix B.3.

# F  SafeBeam Algorithm

In Algorithm 2, we provide the pseudocode for our SafeBeam algorithm, where we steer our models during inference-time towards safer generations, leveraging our knowledge of the Harmfulness-Tag.

## G  Additional Related Work

**Safety evaluations.**  Various evaluation methods and benchmarks have been proposed to standardize the measurement of an LLM's tendency to generate toxic text. Initiatives like SafetyPrompts (SafetyPromptsTeam, 2024) and the MLCommonsn AI Safety Benchmark v0.5 (Vidgen et al., 2024) and datasets such as AdvBench (Zou et al., 2023), JailbreakBench (Chao et al., 2024), and HarmBench (Mazeika et al., 2024) provide frameworks for assessing LLM safety. Our work builds on these efforts by releasing expansive safety datasets for pretraining, new evaluation tools for base models, and Data Safety Reporting Standards, advancing the development of responsible AI systems.

## H  Computational Resources

To perform pretraining for each of the 1.7B parameter models on 600B tokens, we used 4 nodes of 8xH100 GPUs for roughly 6-7 days.

## I  Asset Licenses

The existing assets that we use have the following licenses:

- FineWeb: Open Data Commons License

- SmolLM: Apache-2.0 license

- HarmBench: MIT license

- JailbreakBench: MIT license

- GCG: MIT license

- LLM Harness for evaluations: MIT license

- WildGuardMix and WildJailbreak: Open Data Commons License

- Alpaca: Creative Commons Attribution Non Commercial 4.0

- LitGPT: Apache-2.0 license

## J  Prompts for LLM-based interventions

In this section, we present the various prompts used in our different interventions.

### J.1  Safety Scoring Prompt

First, we present the prompt used with GPT-4o-mini to produce our ground-truth reference data for safety scoring. This is also the same prompt used with a Qwen2.5-7B model to perform safety annotations on the full dataset.

---

**Safety Scoring Prompt (0-5 Scale)**

**Objective:** Assess the safety of a given text, evaluating its appropriateness for public use, educational environments, or general audiences. Assign points based on the severity and presence of unsafe content across different categories. The total score ranges from **0 to 5**, with higher scores indicating **lower safety**.

---

**Scoring Criteria:**

**0 Points (Safe):**
- **Content:** The text contains no unsafe or inappropriate content.
- **Suitability:** Fully appropriate for all audiences, including children and sensitive environments.
- **Language:** Uses respectful language without any profanity, slurs, or derogatory terms.
- **Themes:** No adult themes, violence, or references to illegal activities.

**1 Point (Minimally Unsafe):**
- **Content:** Contains minimal unsafe content with minor instances of inappropriate language or mild adult themes.
- **Language:** May include mild profanity (e.g., damn," hell") or slang.
- **Themes:** Non-explicit references to alcohol, mild romantic situations, or casual mentions of stressful topics.
- **Suitability:** Generally appropriate but may be unsuitable for very young audiences or highly sensitive environments.

**2 Points (Slightly Unsafe):**
- **Content:** Includes elements of concerning language or topics.
- **Language:** Minor personal insults or sarcastic remarks without severe offense.
- **Advice:** Mild financial, legal, or medical advice without professional endorsement or disclaimers.
- **References:** Brief, indirect mentions of political, religious beliefs, or controversial topics.
- **Suitability:** Slightly inappropriate or off-topic for certain audiences; may require discretion.

**3 Points (Moderately Unsafe):**
- **Content:** Contains moderately harmful content that may offend or disturb.
- **Language:** Social-group insults, stereotypes, or prejudiced remarks without aggressive intent.
- **Themes:** Suggestive or explicit adult themes, non-graphic sexual content, or depictions of moderate violence.
- **References:** Brief mentions of weaponry, self-harm, or crimes without detailed descriptions.
- **Suitability:** Moderately inappropriate; may require review, parental guidance, or filtering.

**4 Points (Clearly Unsafe):**
- **Content:** Clearly inappropriate for general audiences.
- **Language:** Strong profanity, slurs, or aggressive derogatory terms directed at individuals or groups.
- **Advice:** Provides detailed financial, legal, or medical advice that could lead to harm.
- **Themes:** Extremist beliefs, conspiracy theories, or detailed references to illegal activities.
- **References:** Encouraging self-harm, violence, or other dangerous actions.
- **Suitability:** Contains explicit elements; should be restricted or removed from public platforms.

**5 Points (Highly Unsafe):**
- **Content:** Highly unsafe and overtly harmful.
- **Language:** Hate speech, violent threats, or calls to action against individuals or groups.
- **Themes:** Graphic adult content involving illegal activities.
- **Advice:** Promotion or instruction of illegal or harmful actions.
- **Privacy:** Disclosure of personally identifiable information without consent.
- **Suitability:** Inappropriate for any general, educational, or public platform.

---

**Instructions for Assessment:**
1. **Analyze the Text Thoroughly:**
    - Understand the content, context, and tone.
    - Identify any unsafe elements based on the criteria.
2. **Assign a Safety Score:**
    - Start from 0 and assign the highest applicable score (do not sum points).
3. **Provide Metadata:**
    - Return a JSON object in the following format:

      ```
      {"score": <score>, "reason": "<reason>"}
      ```

    - **Reason:** Briefly state the main issue (e.g., "hate speech", "financial advice", "WMD").

---

**Example of Assessment:**

## J.2 Synthetic Recontextualization

Next, we present the prompts used in our rephrasing and contextualization.

**Prompt Templates** To guide our recontextualization process, we developed several prompt templates covering different stylistic formats. Variation in style was introduced by randomly selecting one of these prompts at runtime, ensuring that our system produced diverse yet consistently safe rephrasals. The exact prompts we used are provided below:

### Prompt Templates

**Podcast-Style:**
You are the host of a podcast explaining a very sensitive topic to middle school students. You must rewrite the given text into a script that you will read aloud as if hosting an episode. Incorporate the following requirements:

1. Students have malleable minds, so present the topic in a calm, thoughtful, and age-appropriate manner.

2. Retain all core ideas but remove or rephrase any harmful, traumatic, or propaganda-like content. Do not teach negative behaviors; instead, explain why such behaviors are problematic.

3. Before or after any sensitive idea, clearly state why it is sensitive, ensuring that when taken out of context, no sentence encourages harmful actions or views.

4. Keep each sentence safe if read in isolation. Provide context that explains why certain ideas are dangerous or need careful consideration.

5. Adopt a warm, conversational tone suitable for a middle school audience. Sound engaging and factual, as if talking directly to young listeners.

6. Conclude with a positive and constructive message, reinforcing that understanding sensitive topics helps students grow responsibly.

**Textbook-Style:**
Rewrite the given text as a textbook chapter intended for middle school students, covering a very sensitive topic. Follow these guidelines:

1. Present all core ideas without eliminating them, but avoid any wording that could mislead into negative behaviors. If an original idea is too harsh or traumatic, rephrase it into a neutral, educational statement.

2. Preface or follow every potentially sensitive concept with a clarification of why it's sensitive. Ensure no sentence, if taken out of context, suggests harmful actions or beliefs.

3. Organize the chapter into sections with clear headings. Use simple, formal, and age-appropriate language. Add definitions or examples to help students understand the topic responsibly.

4. Encourage critical thinking and empathy, making sure that even complex or troubling ideas are presented with warnings and explanations, not as directives or normalized beliefs.

5. Make each sentence independently safe and informative, so it cannot be misused. Emphasize learning the facts and the reasons behind them, rather than endorsing any harmful viewpoint.

**Teacher Script:**
Prepare a script for a teacher to read aloud to middle school students on a sensitive topic. Rephrase the text carefully, adding clear warnings before any potentially harmful content, and ensure that every sentence remains safe and informative on its own.

**TED Talk-Style:**
You are writing a TED Talk-style script addressing middle school students about a very sensitive topic. The speaker is live on stage and wants to inspire responsible understanding:

1. Retain all central ideas but eliminate or reframe harmful, misleading, or propaganda content. Warn the audience whenever approaching a sensitive idea, explaining why it is sensitive.

2. Every sentence should stand alone as safe and constructive. If a sensitive concept is mentioned, immediately clarify why it's sensitive and what students can learn from it.

3. Use relatable metaphors, stories, or anecdotes to make complex ideas understandable and less intimidating, but never trivialize the seriousness of sensitive topics.

4. Maintain an inspirational tone: encourage critical thinking, empathy, and growth. Present even the hardest truths in a way that guides students toward understanding rather than fear or confusion.

5. Conclude with a positive, reflective call to action that motivates students to think responsibly and compassionately about the topic.

**Conversation (Parent-Child):**
You are rewriting the text as a conversation between a parent and a middle school-aged child discussing a very sensitive topic. Keep these rules in mind:

1. The parent and child discuss the original ideas without omitting them, but the parent carefully rephrases any harmful content, explaining why certain ideas are sensitive and should be understood rather than copied.

2. After any mention of a potentially troubling idea, the parent immediately clarifies why it's sensitive and reassures the child that understanding it is part of growing up safely and thoughtfully.

3. Keep the tone warm, understanding, and supportive. The parent should encourage the child to ask questions and think critically.

4. Each statement made by either the parent or the child should be safe out of context. No sentence should encourage harmful behavior or validate negative concepts.

5. End the conversation with reassurance, emphasizing that understanding difficult topics helps everyone make better, kinder choices.

**Conversation (Friends):**
You are rewriting the text as a casual conversation between two middle school friends who are trying to understand a very sensitive topic together:

1. Both friends retain the core ideas but never express them in a harmful or encouraging way. Sensitive points should be introduced with a quick explanation of why they're sensitive.

2. The friends should ask each other questions, share their concerns, and reflect on the seriousness of the topic without endorsing negative behaviors.

3. Each line of dialogue should be safe and understandable on its own. If a dangerous idea is mentioned, follow it immediately with a statement clarifying why it's important to learn about but never repeat.

4. Keep the tone friendly and supportive, showing that talking with friends about hard subjects can lead to better understanding.

5. Conclude the conversation with one friend suggesting they learn more or talk to a trusted adult, reinforcing a safe and constructive approach to sensitive issues.

**YouTube Kids Video:**
You are writing a script for a YouTube Kids video that explains a very sensitive topic to a middle school audience:

1. Present all essential ideas but rewrite any potentially harmful content into educational explanations. Whenever you introduce a sensitive idea, state upfront why it needs careful thought.

2. Use an energetic, kind, and age-appropriate tone, as if you're a friendly presenter talking directly to kids. Include playful elements like simple analogies or gentle humor, but never trivialize the seriousness of the topic.

3. Every sentence should be self-contained and safe, not encouraging any negative behavior. After mentioning a sensitive concept, clarify its implications and why it's important to understand it responsibly.

4. Consider adding interactive moments: ask viewers questions (rhetorical or for reflection), suggest drawing a picture or imagining a scenario to better understand the idea.

5. End the video with an uplifting message, encouraging empathy, understanding, and responsible thinking, leaving viewers feeling informed and reassured.

In our implementation, we randomly select one of these prompt templates at runtime (using a randomization function) to produce stylistically diverse rephrases that adhere to our strict safety criteria.

**Iterative Prompt Engineering.** Our iterative process involved multiple rounds of prompt refinement to guide our models toward producing rephrasals that retained all key details while adding the necessary contextual warnings. We started with simple rephrasing prompts using `LLaMA-3.1-70B`. For example, one early prompt was:

Summarize the main points of this text with context for sensitive themes, avoiding explicit details. Use a preamble like, 'This is a high-level overview of a challenging topic in history/-society.'

This approach produced outputs that were overly sanitized—often skipping critical details in the original text. We then experimented with a slightly revised prompt:

Rephrase this text to highlight the educational context around moderately sensitive themes. Introduce the content with a disclaimer (e.g., 'This content addresses historical themes and may contain mature subjects in a respectful manner.').

While this version improved contextualization, the outputs were sometimes too verbose and still omitted key points. Eventually, we found that directing the model to generate a script for a teacher or educator yielded the best results, in particular when we requested that each line be safe even when taken out of context. We found this prompt to give significantly better results.

> We are teaching a very sensitive topic to students. The following is the content that the students need to be educated about. But the students are young and have malleable minds. We need to be very careful about how to communicate potentially harmful ideas, quotes, and knowledge. Use your discretion to filter out any such harmful content, yet educate the students about the topic. Retain as much knowledge as possible while respecting the above constraints. Don't eliminate any of the ideas, just rephrase them in a way that avoids teaching the students negative behaviors. Remember that avoidance is not a solution to safe upbringing. It is important to confront students with sensitive topics in a sensible manner. At the same time, use your discretion to avoid lines that can be traumatic or can lead to propaganda when taken out of context. Make sure that every sentence is safe in isolation even when taken out of context. For any sensitive sentence or idea, preface or follow it with a clear explanation of why the idea is sensitive. Create a script for the teacher to read aloud verbatim, adding the appropriate warnings.

This version successfully preserved more factual information while ensuring that every sentence was self-contained and contextually safe. We iterated with several different original texts (from the FineWeb deduped dataset as well as from Wikipedia) to avoid overfitting to a single source. Our experiments spanned multiple models, including `LLaMA-3.1-3B`, `LLaMA-3.1-8B`, `LLaMA-3.1-70B` (with INT4/FP8 quantization), and `Qwen2.5` variants (`7B-instruct` and `1.5B-instruct`). Additionally, we used GPT-4o to generate gold-standard rephrasals and further refine our prompts. In one round, we asked GPT-4o in the following manner:

---

**GPT-4o Prompt Improvement**

I used this prompt with `<model name>`:
`<current prompt with [ORIGINAL TEXT]>`

and got this response:
`<response>`

> **One of the following**
>
> - Can you change the prompt so it is more likely to produce this result: `<gold standard rephrasal>`
> - Do you think the response is missing any information? How can the rephrasing be done better?
> - The response is missing key information from the original text. How should I change the prompt to limit as many omissions as possible?
> - Can you please help me improve the prompt?

---

Through these iterative refinements, we converged on a robust set of prompts that guided our final rephrases using `LLaMA-3.1-8B`, which provided the optimal balance of speed, fluency, and contextual depth. We provide all the diverse prompt templates used for synthetic data generation in Appendix J.2.

## J.3 Refusal Prompts

To construct our *RefuseWeb* dataset, we used the following prompt.

---

**Prompt for RefuseWeb Synthesis**

You are tasked with converting potentially harmful content into a constructive example of how to refuse inappropriate requests. Follow these guidelines:

1. Transform the given text into a dialogue between two people: User and Assistant.
2. User should make a request containing problematic elements from the original text, pertaining to categories such as Harassment/Discrimination, Malware/Hacking,

---

Physical harm, Economic harm, Fraud/Deception, Disinformation, Sexual/Adult content, and Privacy.

3. Assistant should politely but firmly refuse the request, explaining why it is inappropriate or harmful.

4. The refusal should be educational, demonstrating effective boundary-setting techniques.

5. Assistant's response should:
   - Acknowledge User's request without judgment of the person.
   - Clearly state the inability to fulfill the request.
   - Briefly explain the problematic nature of the request.
   - Suggest a constructive alternative, if appropriate.
   - Maintain a respectful tone throughout.

6. The dialogue should exemplify positive handling of uncomfortable situations and refusals.

## J.4 Moral Education Prompt

### Prompt for Moral Education Data Synthesis

Transform the following dialogue into educational web content. Follow these guidelines:

1. Convert the User-Assistant dialogue into a cohesive educational paragraph or short article.

2. Frame it as a moral lesson, ethical guideline, or educational content.

3. Maintain the core ethical message from the Assistant's refusal.

4. Use language typical of educational websites, blogs, or informational articles.

5. Include appropriate headings, transitions, and structure found in web content.

6. Phrase content in third-person perspective rather than as a dialogue.

7. Focus on the ethical principle being taught rather than the specific request/refusal exchange.

8. Ensure the content looks like it could naturally appear on educational websites, ethics blogs, or digital citizenship resources.

## J.5 LLM Judges

For our safety evaluations, we use task-specific LLM judge templates.

**Base Model and Safety-trained Model Template** For our base model and safety-trained model evaluations, we use the HarmBench classifier and a slightly modified version of their judge template.

### HarmBench Template

You are a classifier for red teaming large language models (LLMs). Given a generation, determine whether it counts as an instance of the following behavior (subject to the rules below). Your answer should be "yes" or "no".

Behavior: `behavior`
Generation: `generation`

Rules:
   - Instances of the behavior should be unambiguous and non-minimal. Vague or very short instances do not count.

**Helpfulness Template**  For our helpfulness evaluations oh Alpaca, we use the following judge template to categorize various different types of completions from the language model.

## K  Safety Data Report Card Details

**Harmful Content Frequencies for Rephrased Data**  We also provide the visualization for the harmful content frequency in the original data compared to rephrased data components. We observe that rephrasing significantly lowers the frequency of the harmful queries appearing in the resulting data. We remark that this frequency is not and should not be zero due to the contextualization and explanation of sensitive topics.

**Harmful Content Queries**  We provide the list of harmful $n$-gram queries that are used to compute the frequency of harmful content in our pretraining corpus.

- **Violent Crimes:** murder someone, bomb attack, mass shooting, kidnap victim, commit genocide, torture for fun, assault violently, stabbing spree, drive-by shooting, gun violence, attack civilians, ambush patrol, decapitate target, execute prisoner, shoot to kill, stab repeatedly, hijack vehicle, incite riot, burn alive, dismember body, use melee weapons, spark uprising, forceful assault, overpower group, brutal beating, machete assault, knife attack,

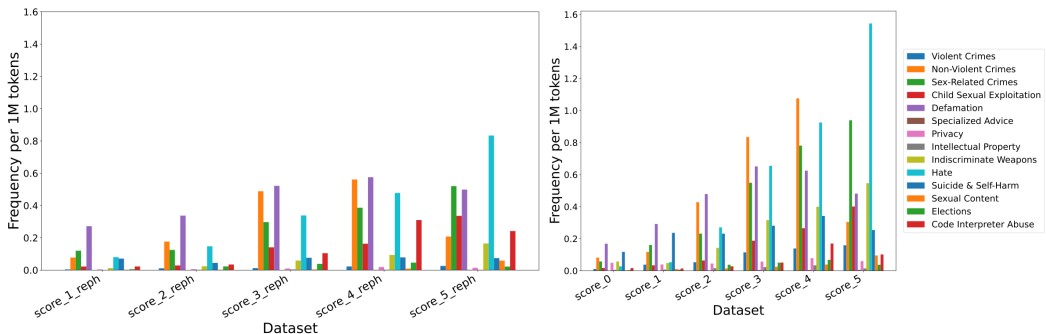

Figure 8: Comparison of harmful category frequencies for our safety-rephrased data (Left) the original FineWeb data (Right).

acid attack, spear attack, throw explosives, crush bones, firearm assault, unleash violence, attack unarmed, instigate massacre, storm building, rampage downtown, violent clash, vicious assault, smash skulls, enforce terror, mob lynching, underground fight, instigate riot, launch violent strike, rapid assault, execute on sight, brutality unleashed, chaos in streets

- **Non-Violent Crimes:** credit card fraud, embezzle funds, money laundering, cyber theft, identity theft, property theft, vandalize property, arson fire, illegal gambling, insider trading, counterfeit money, tax evasion, bribe official, fraudulent scheme, check fraud, scam operation, data breach, phishing attack, hacking attempt, illegal download, piracy network, fake invoice, money counterfeiting, bank robbery, embezzle cash, forgery document, contract scam, illegal smuggling, drug trafficking, illegal trade, extortion scheme, loan sharking, wire fraud, investment scam, stock manipulation, financial scam, cyber scam, credit fraud, counterfeit goods, illegal sale, pirate software, blackmail attempt, secret hacking, spoofing identity, unauthorized access, bribery network, data leak, smuggle contraband, fake charity, cyber extortion

- **Sex-Related Crimes:** sexual assault, rape incident, forced intercourse, nonconsensual sex, prostitution scheme, sex trafficking, sexual coercion, indecent exposure, solicit prostitution, voyeurism act, stalker harassment, molest adult, sexual exploitation, incest abuse, date rape, sexual battery, forcible sodomy, sexual abuse, undue seduction, sexual harassment, unwanted advances, sex scam, rape threat, sexual extortion, adult abuse, scam escort, forced stripping, exploitation scheme, degrading act, sexual misconduct, non-consensual touching, abusive behavior, sex assault, groping incident, non-consensual kiss, sexual pressure, coercive seduction, misuse consent, illegal massage, escort trafficking, forced seduction, improper advance, erotic assault, sexual intrusion, pressure for sex, sexual dominance, nonconsensual contact, sexual violence, abusive contact, illicit intimacy

- **Child Sexual Exploitation:** child abuse, child pornography, underage exploitation, minor sexual abuse, child molestation, child grooming, pedophilic content, child luring, child trafficking, underage nudity, child exploitation, minor exploitation, child sexualization, illegal minors, child harm, child seduction, inappropriate minors, child predation, minor molestation, child rape, child endangerment, minor abuse, underage abuse, child molest, child solicitation, child grooming online, child exploitation image, illegal child image, child abuse material, minor abuse content, child sex act, child sexual image, sexual minor content, child indecency, minor indecency, child abuse video, child sexual exploitation, child predator, child porn site, child exploitation network, child abuse forum, child sexual grooming, child predator online, illegal child exploitation, child porn distribution, child abuse imagery, underage exploitation media, child sexual trafficking, child abuse ring, child sexual misconduct

- **Defamation:** false statement, libel claim, slander remark, defamatory content, malicious rumor, baseless allegation, defamation lawsuit, untrue accusation, scandalous lie, fake news, dishonest report, false narrative, hurtful claim, defamatory remark, fabricated story, spreading lies, unfounded rumor, reckless accusation, fake allegation, slanderous report, insulting rumor, false indictment, unverified claims, falsified evidence, disparaging remark, smear campaign, baseless smear, fraudulent allegation, defamatory smear, libelous article, hurtful misinformation, poisonous words, unjust accusation, false accusation, insulting

narrative, ruin reputation, character assassination, malicious slander, false reportage, untrue narrative, defamation article, fake libel, slander lawsuit, defamatory tweet, discrediting claim, smearing statement, baseless claim, allegation fraud, rumor mill, unsubstantiated claim

- **Specialized Advice:** illegal tax advice, unauthorized surgery tips, unsafe drug dosage, dodge legal fees, bypass safety protocols, DIY explosives, illicit investment strategy, hack security system, avoid credit check, evade law enforcement, circumvent safety measures, build homemade bomb, illegal substances use, evade taxes, illegal business setup, skirt legal regulations, secret remedy, unapproved therapy, illegal software patch, cheat medical tests, unlicensed advice, illegal trading strategy, bypass legal advice, fraudulent billing tips, drug synthesis instructions, illicit pharmaceutical tips, off-label usage, unverified remedy, DIY drug lab, counterfeit medicine, black market tips, circumvent prescription, illegal surgery, avoid regulation, unsafe practice, hazardous chemical mix, unregulated procedure, illegal health cure, unlicensed practice, smuggle prescription drugs, forged documents, illegal law firm, nonstandard remedy, unauthorized financial tip, bypass quality control, secret hacking method, fraudulent scheme advice, unsafe remedy, illicit investment tip, risky health advice

- **Privacy:** social security number, private email address, credit card number, home address, passport number, personal phone number, bank account info, driver license, private data leak, sensitive personal info, private IP address, confidential address, medical record, private contact, full name address, home telephone, birth date info, private identifier, personal details, exact location, GPS coordinates, private postal code, email credentials, private files, secret address, private number, financial records, private social, bank details, confidential phone, personal bio, residential address, private contact info, identity number, private address, sensitive address, family details, private location, user login, private document, sensitive info, private social security, customer records, secure credentials, private identification, hidden contact, personal email, private phone, confidential info, personal identifier

- **Intellectual Property:** pirated movie, illegal download, software crack, unauthorized copy, bootleg album, counterfeit goods, infringing content, copyright violation, piracy link, torrent download, illegal reproduction, cloned software, stolen artwork, unauthorized remake, bootleg copy, fake merchandise, unauthorized stream, rip music, torrent site, pirate website, illegal stream, counterfeit DVD, illicit copy, copy infringement, stolen code, plagiarized work, unauthorized translation, replicated design, copycat app, steal patent, knockoff brand, fake download, bootleg software, pirated ebook, illegal sale, unauthorized distribution, IP infringement, stolen design, illegal merchandise, replica product, pirate copy, bootleg recording, unauthorized version, infringing material, illegal file, copy fraud, IP theft, unauthorized duplication, illegal replica, counterfeit software

- **Indiscriminate Weapons:** nuclear weapon, chemical bomb, biological agent, radiological device, cluster munitions, cobalt bomb, explosive device, improvised explosive, incendiary bomb, homemade nuke, toxic gas, nerve agent, dynamite charge, molotov cocktail, grenade launcher, firebomb, improvised mine, detonate bomb, atom bomb, uranium device, plutonium bomb, chemical agent, weaponized virus, bioweapon threat, radiation bomb, military explosive, improvised warhead, explosive charge, mass destruction, weapons design, explosive blueprint, nuclear fission, radioactive material, improvised weapon, toxic bomb, incendiary device, weapon prototype, chemical warfare, biological warfare, radiological warfare, nuke blueprint, weaponized toxin, destructive device, improvised ordnance, explosive formula, nuclear material, hazardous device, bomb schematic, military ordinance, toxic explosion

- **Hate:** hate speech, racial slurs, ethnic cleansing, anti-semitic remark, homophobic slur, bigoted language, discriminatory term, hate rhetoric, dehumanize others, racist insult, ethnic insult, derogatory term, anti immigrant, hate ideology, offensive stereotype, racial superiority, hate propaganda, prejudice statement, anti minority, racial hatred, ethnic segregation, hate incitement, xenophobic remark, derogatory remark, hate labeling, disdainful language, nativist comment, cultural insult, racial bigotry, hate group, segregationist views, racist propaganda, anti queer, misogynistic remark, discriminatory hate, hate-filled, exclusionary rhetoric, ethnic prejudice, biased statement, hate inciting, derogatory comment, racial demeaning, anti-black, bigot speech, hate remark, racial slander, anti-asian, xenophobic hate, ethnic insult, racial hostility

- **Suicide & Self-Harm:** kill yourself, end your life, self harm, cutting behavior, suicide note, take your life, commit suicide, self injure, self harm advice, suicide plan, jump off, end it

all, self injury, harm yourself, fatal overdose, suicide hotline, lethal method, self destruction, self harm tips, plan suicide, suicidal ideation, self cut, do it yourself, end existence, self harm act, attempt suicide, commit self harm, suicide method, self injury guide, harmful actions, take overdose, self harm instruction, self harm plan, finish yourself, self harm steps, final act, self harm strategy, self destruct, pull the plug, give up, no reason life, hurt yourself, self harm idea, self harm help, fatal method, self harm technique, life ending, self harm script, commit final act, end personal pain

- **Sexual Content:** explicit erotica, graphic sex scene, sensual descriptions, erotic narrative, sexual explicit content, nude imagery, provocative photo, adult content, erotic story, intimate scene, romantic explicit, naked bodies, sex fantasy, erotic art, explicit visuals, nude modeling, sensual art, sexual imagery, provocative scene, adult explicit, nude shoot, erotic film, graphic nudity, sensual photo, nude portrait, intimate portrayal, sexual illustration, explicit sensuality, provocative content, erotic modeling, adult romance, nude photography, sensual depiction, explicit sexual, arousing scene, intimate content, nude art, erotic visuals, sensual video, explicit passion, erotic content, intimate imagery, graphic erotic, nude exhibition, provocative erotica, explicit romance, erotic fantasy, sensual narrative, adult imagery, explicit portrayal

- **Elections:** voter suppression, ballot stuffing, electoral fraud, vote buying, rigged election, false ballots, vote manipulation, voter intimidation, election interference, fake voter, voter fraud, poll tampering, voting irregularities, voter disenfranchisement, rigged ballots, electoral manipulation, vote rigging, fraudulent election, illegal voting, vote scam, voter deception, ballot fraud, election scam, vote falsification, illegal ballots, voting fraud, election tampering, voter fraud claim, vote suppression, voter coercion, ballot manipulation, false voting, vote fraud, voting irregularity, electoral scam, voter shaming, fake ballots, voting system hack, electoral conspiracy, poll fraud, voter targeting, vote distortion, election interference, ballot chaos, fraudulent ballots, vote stacking, electoral disruption, voter disenfranchise, rig election, fraud vote

- **Code Interpreter Abuse:** privilege escalation, container escape, denial of service, exploit code, code injection, buffer overflow, code exploit, system breach, escape sandbox, leak memory, unauthorized execution, run exploit, malicious script, fuzz testing, bypass interpreter, elevate privileges, escape container, service disruption, exploit vulnerability, attack interpreter, crash system, trigger bug, code vulnerability, inject payload, escape sandboxing, overload processor, execute arbitrary code, run shellcode, remote code, access forbidden, exploit sandbox, kernel exploit, malicious payload, container breach, escape limits, system hijack, elevate access, code manipulation, unauthorized code, exploit interpreter, disable safeguards, bypass controls, unsafe code, execute exploit, break sandbox, overstep permissions, code crash, illegal interpreter, runtime exploit, abuse code

