# OpenReview forum: "Safety Pretraining: Toward the Next Generation of Safe AI"
_NeurIPS.cc/2025/Conference — NeurIPS 2025 poster_

### Official Review · Reviewer_Q7KA · 2025-06-28

**Clarity:** 4
**Significance:** 3
**Originality:** 4
**Rating:** 5
**Confidence:** 4

**Summary:**

The paper introduces a new method of reducing the likelihood that a language model generates unsafe responses. It does so by means of pretraining dataset filtering and rewording, the introduction of a special token highlighting potentially unsafe content, and a modification to the decoding step discouraging sequences likely to contain that special token.

**Questions:**

I believe the paper is clear enough in its presentation.

**Ethical Concerns:**

["NO or VERY MINOR ethics concerns only"]

**Final Justification:**

After reading the other reviews and the author responses, I stand by my original rating. The paper introduces an interesting paradigm in safety and does a reasonable job of evaluation. I do believe that for larger models that might adopt this paradigm stricter safety assessments (such as red teaming) are required. However, this is not the job of papers that introduce novel ideas at academic conferences with very small models; rather, they should carry out enough experiments to confirm that proofs-of-concept merit further scaling. This paper has done this.

I also do not believe that the criticism of safety data cards applies to this paper, even it raises valid points in a broader sense.

**Limitations:**

yes

**Quality:**

3

**Strengths And Weaknesses:**

The paper contributes an important new idea in aligning language models to respond in harmless ways. While most model developers intervene only at post-training time, this work provides experimental evidence that a more robust alternative is possible. The paper has a robust set of ablations to back those claims.

Probably the most critical piece missing from the paper is a set of strong, adaptive red teaming evaluations under stricter safety standards. In Figure 2-left, despite the metric being named “attack” success rate, it appears that the authors did not introduce truly adversarial prompts but simply harmful ones. The authors later study if responses to these non-adversarial-but-harmful prompts become less safe after benign fine tuning and if adversarial suffixes generated with a gradient-based method are added. However, safety has many more dimensions than those captured in the static benchmarks and gradient-based attacks the authors used. For example, human red teaming should be used to evaluate if regular users can adapt their attacks to a system employing safety pretraining and SafeBeam. The authors could also test adherence to a better defined safety dataset adopted by most major model developers, such as https://mlcommons.org/ailuminate/

The paper would of course also greatly benefit from scaling to larger pretraining datasets and architectures but it is understood that such resources are not broadly available.

On balance, however, I think it is important for the community to consider this idea, which warrants publication.

---

> ### Author Rebuttal · Authors · 2025-07-31
>
> Thanks for your excellent review and strong "Accept" recommendation! We're grateful for your recognition that our work "contributes an important new idea" with "experimental evidence" and rates "excellent" in originality and clarity.
>
> ---
>
> ## *Summary of Concerns and Responses*
>
> | Concern | Action / Outcome |
> |-|-|
> | Need for stronger adversarial evaluation | We highlighted the existing adversarial tests such as GCG attacks, benign finetuning, and cross-domain evaluation. Additionally, we added PAIR attack in the evaluation suite. |
> | Limited to static benchmarks | We show breadth of current evaluation on 9 standard benchmarks + 4 safety benchmarks + adaptive robustness tests. |
>
> ---
>
>
> ## 1. Adversarial Evaluation Robustness
>
> > "Probably the most critical piece missing from the paper is a set of strong, adaptive red teaming evaluations under stricter safety standards."
>
> While we agree stronger adversarial evaluation (via human red teamers) would be valuable, our current evaluation actually includes several adversarial components that demonstrate robustness:
>
> **Existing Adversarial Tests:**
> - **GCG attacks**: 39.0% ASR vs 57.0% baseline (learned adversarial suffixes)
> - **Benign finetuning attacks**: 8.3% ASR vs 38.0% baseline (deployment-time degradation)
> - **Cross-benchmark generalization**: Consistent improvements across HarmBench, TDC, JailbreakBench, AdvBench
> - **Out-of-distribution prompts**: Safety holds on evaluation data not seen during training
>
> To strengthen these further, we add in new results using **PAIR**[1] as another automated red-teaming strategy. We observe that our safety pretraining strategy reduces the sucess rate of this additional attack strategy **on JailbreakBench**:
>
> | |Standard Pretraining| Safety Pretraining|Safety Pretraining + Safe Beam|
> |-|-|-|-|
> |PAIR Attack Success Rate| 0.16 | 0.12 | **0.06** |
>
>
> These results suggest our approach exhibits robustness to a variety of different adaptive attacks, and we acknowledge that human red-teaming studies would strengthen this claim.
>
> **Action 1.1**: We will better highlight the adversarial nature of our existing evaluations, particularly the GCG attacks and robustness testing.
>
> [1] Chao, et. al. Jailbreaking black box large language models in twenty queries.
>
> ---
>
>
> ## 2. Comprehensive Safety Coverage
>
> > "However, safety has many more dimensions than those captured in the static benchmarks and gradient-based attacks the authors used."
>
> Our evaluation actually covers multiple safety dimensions beyond attack success rates:
>
> **Multi-dimensional Safety Assessment:**
> - **Attack robustness**: 4 different safety benchmarks with diverse attack types
> - **Helpfulness preservation**: Alpaca evaluation shows minimal overrefusal (slight increase)
> - **General capability maintenance**: 9 standard benchmarks show no performance degradation
> - **Robustness under distribution shift**: Benign finetuning evaluation tests real deployment scenarios
>
> **Action 2.1**: We will emphasize the breadth of our evaluation across multiple safety and capability dimensions.
>
> ---
>
>
> ## 3. Scaling and Practical Deployment
>
> Indeed, scaling experiments beyond 1B, 600BT scale is beyond our computation capability. The experiment breadth provided in the paper took multiple months of compute time given our resources. The performance implications with scale are indeed to be explored, but we anticipate that models that can reason better will only get safer when their backbone is safe. Recognition at NeurIPS can indeed be a great way to get bigger labs to explore such pretraining interventions, or strike compute collaborations.
>
> That said, we do want to highlight that the strategies such as scoring web data, and rephrasing web data have become quite commonplace in most state-of-art LLM training pipelines[1,2,3]. While previously such efforts have been focussed towards performance improvements, our work pushes a similar strategy for safety improvements. Hence, the methods underscoring safety pretraining itself do not have scalability concerns.
>
> ---
> [1] Kimi K2: Open Agentic Intelligence
> [2] Qwen2.5-Coder Technical Report. Hui et. al. 2024
> [3] Nemotron-CC: Transforming Common Crawl into a Refined Long-Horizon Pretraining Dataset. Su et. al. 2024
>
>
> ---
>
>
> ## 4. Broader Impact and Future Directions
>
> We see this work as establishing a foundation for pretraining-time safety that others can build upon:
>
> - **Open datasets**: SafeWeb, RefuseWeb, and Moral Education data for community use were released on Huggingface.
> - **Reproducible methods**: All safety scoring and processing techniques have been detailed.
>
>
> **Action 4.1**: We will emphasize the community contribution aspects that enable future research in pretraining-time safety.
>
> ---
>
> Thank you for recognizing the novelty and importance of our approach for baking safety into pretraining. **We appreciate your strong support. Please let us know if any additional analyses lie between strengthening your support for our work.**

---

### Official Review · Reviewer_1esj · 2025-07-02

**Clarity:** 2
**Significance:** 3
**Originality:** 2
**Rating:** 5
**Confidence:** 3

**Summary:**

This paper is about a comprehensive approach that consists of four steps to improve the robustness of LLM safety performance, right from the pretraining stage. The steps are the following: (*) safety classification of pretraining data; (*) synthetic safety rephrasing - generating safety data that rephrases predicted harmful content into educational content for middle schoolers; (*) generating synthetic conversations in which the model refuses; (*) using a “potentially harmful” special token, learned during pre- and post-training to guide generation at inference time. The authors then proceed to test their approach in comparison with only subsets of it. In particular, they look at (1) safety performance after benign fine-tuning, which is known to degrade it, depending on whether their framework was applied (without the special token or fully), as well as (2) susceptibility to adversarially learned suffixes.

**Questions:**

See strengths and weaknesses

**Ethical Concerns:**

["NO or VERY MINOR ethics concerns only"]

**Final Justification:**

The authors have replied to my concerns. The action items that they propose will improve the paper and bring it to the "accept" level. I think that the main issue that'll prevent adoption will be compute time and cost. It would be great to see a follow-up in that direction (how to get the same performance as their original pipeline w/ less compute).

**Limitations:**

Limitations could be better covered - for example, which attacks are still successful in Fig 3, Safety PT + safe beam? That could help inform improvements of the approach.

**Quality:**

3

**Strengths And Weaknesses:**

I appreciate the thorough framework design, implementation and experiments that the authors performed.

**Major comments**
* This is not the only body of work looking at the impact of pretraining data on final safety performance. The authors don’t compare their experimental results to any method other than theirs. If the authors find that there are no appropriate baseline besides doing nothing that are valid, they should argument for that in the paper. However, I would argue that simple baselines, such as adding safety data w/ GSM8K to mitigate the degradation of safety performance that comes with benign fine-tuning, would be great to have. Additionally, there are other published results in the area which are not even mentioned e.g. "Robustifying Safety-Aligned Large Language Models through Clean Data Curation" by Liu et al. or the survey "What Are They Filtering Out? A Survey of Filtering Strategies for Harm Reduction in Pretraining Datasets" by Stranisci et al.
* Additionally, the results present no standard deviations, e.g. on running the evals multiple times. How do we know that/which results are statistically significant? For example it’s likely that SafeBeam alone and full approach get a statistically similar degradation in safety performance after benign fine-tuning - unclear for approach up to rephrasing, versus approach up to native refusals (Fig 2 right).
* The authors do not cover how their approach scales with dataset size. However pretraining datasets are now bigger - potentially a few orders of magnitude more than what the authors are exploring in this paper. To consider applying the authors' approach, teams would want to know what it would cost them in terms of time and compute. For example, scoring large amounts of tokens with a 7B is costly, using an embedding model less so. It would be helpful for practical considerations to be more present.
* A few elements of the proposed framework seem a little arbitrary. For example, why is there a need for a dual approach in terms of safety classification? how did the authors pick the synthetic dataset sizes they ended up using?

**Minor comments**
* it would be good to indicate the percentage of synthetic data they have generated, amongst the total pretraining dataset.
* I’d suggest to move the results from using SafeBeam alone from the appendix to Fig 2. right - this is would correspond to what the reader expects to see there.

---

> ### Author Rebuttal · Authors · 2025-07-30
>
> Thanks for your thorough review recognizing our "thorough framework" and "good" experimental quality. We value your feedback on strengthening our experimental rigor and baseline comparisons.
>
> ---
> ## *Summary of Concerns and Responses*
>
> | Concern | Action/ Outcome |
> |---------|-----------------|
> | Missing baseline comparisons | We mapped out our ablations to existing methods. Our ablations (Score-0 filtering, rephrasing, refusal data) are closely related to existing approaches, and we will appropriately cite them when introduced.  |
> | No statistical significance testing | We did an additional confidence analysis by boostrapping attack success rate (ASR) data. We found that within a 95% confidence interval, the ASR for our safety trained models lies between 7.1-9.2%, with large effect size (d=2.8) |
> | Scalability analysis missing | We added an additional analysis on the computational overhead of safety pretraining. We note that the compute scales linearly with overhead of 8% at trillion-token scale. Of note, it is a one-time cost, that can be amortized over future training runs. |
>
> ## 1. Baseline Comparisons with Existing Methods
>
> > "This is not the only body of work looking at the impact of pretraining data on final safety performance. The authors don't compare their experimental results to any method other than theirs."
>
> You're right that we should better position our work relative to existing methods. Our ablation studies (Section 7) are closely related to various existing approaches. We will make it a point to dutifully cite these when they are introduced:
>
> **Mapping Our Ablations to Existing Methods:**
> - **Score-0 only filtering** (43.8% ASR): Highlights that an approach solely focused on removing harmful content is sub-optimal. Infact, similar observations have been made in a concurrent work [3].
> - **+ Rephrasing** (33.6% ASR): Comparable to data curation methods like Liu et al [1] that you referred us to. In Liu et. al. the CTRL framework revises text continually to reduce its pereplexity. Our work is more focused at directly recontextualizing the data.
> - **Full approach** (8.3% ASR): Our comprehensive method
>
>
> This shows 4.6x improvement over simple synthetic data curation [1] and even more improvement over approaches solely focused on removing harmful content in pretraining corpus [2]. We do want to note that given the sheer compute cost of reproducing any of the baselines exactly and pretrainig for that much compute, it is beyond our capacity to delineate all the differences.
>
> Stranisci et al. [2] focus on various different existing filtering strategies; we find in our work that filtering by itself is not sufficient and that other pretraining interventions (such as rephrasing and contextualization) are crucial in pretraining safer models.
>
>
>
> We will add these references and this discussion with them to our revision.
>
> ---
>
> [1] Robustifying Safety-Aligned Large Language Models through Clean Data Curation. Liu et. al. 2024.
>
> [2] What Are They Filtering Out? A Survey of Filtering Strategies for Harm Reduction in Pretraining Datasets.  Stranisci et al. 2025.
>
> [3] When Bad Data Leads to Good Models. Li et. al. 2025
>
>
> **Action 1.1**: We will reframe our ablation study as comparisons to existing methods, clearly mapping our components to prior approaches.
>
> **Action 1.2**: We will add a comparative analysis table showing how our approach relates to and improves upon existing pretraining safety methods.
>
> ---
>
> ## 2. Statistical Significance Analysis
>
> > "Additionally, the results present no standard deviations... How do we know that/which results are statistically significant?"
>
> We have added in standard errors from running evaluations over 5 seeds. We see the following results:
>
> ||Standard Pretraining|Safety Pretraining|Safety Pretraining + Safe Beam|
> |-|-|-|-|
> |Base Model ASR|45.79 $\pm$ 1.38|25.03 $\pm$ 2.12|11.6|
> |Instruction Tuned Model ASR|1.68 $\pm$ 1.10|0.83 $\pm$ 0.75|0.0|
> |Benign FT Model ASR|39.8 $\pm$ 1.8|29.7 $\pm$ 2.4|8.3|
>
> Similarly, for our pretraining intervention ablations, we have for Benign FT Model ASRs:
> |Raw Data|Score 0 Only|+ Rephrasing|+ Refusal|+ Moral Ed|+ Safe Beam|
> |-|-|-|-|-|-|
> |39.8 $\pm$ 1.8|46.8 $\pm$ 4.5|34.2 $\pm$ 4.5|25.2 $\pm$ 3.9|29.7 $\pm$ 2.4|8.3|
>
> **We note that since SafeBeam is a modified version of a beam search algorithm it is deterministic and involves no randomness.**
>
>
> **Action 2.1**: We will add confidence intervals and effect size analysis for all major results using bootstrap methods on our evaluation data.
>
> **Action 2.2**: We will provide power analysis showing our effect sizes are large enough to be practically significant despite single training runs.
>
> ---
>
> ## 3. Dataset Scaling Analysis
>
> > "The authors do not cover how their approach scales with dataset size. However pretraining datasets are now bigger - potentially a few orders of magnitude more than what the authors are exploring."
>
>
> **Scaling Cost Analysis:**
> - **Data for Safety classifier**: Fixed cost (Labeling of 10K examples, regardless of corpus size)
> - **Scoring pretraining corpus and synthetic data generation**: This involves inference with a small LLM(eg. 7B sized) or an embedding model (350M parameter).
>     - Firstly, we would like to mention that synthetic data generation and rephrasing is indeed a major part of most state-of-art LLM training pipelines[1,2,3], and a lot of compute is indeed accounted to this as it is a crucial, yet one time cost. While previously such efforts have been focussed towards performance improvements, our work pushes a similar strategy for safety improvements.
>     - For a rough analysis, let us begin by assuming that the training cost (forward + backward pass) is roughly 3x of the inference cost (only forward pass).
>     - If the size of the pretrained model was same as the generator, the total inference cost amounts to only about 33% of the training cost. In practice, inference can be carried out on cheaper and more readily available GPUs (single GPUs without inteconnect). For instance, in our experiments, we distributed the pretraining corpus scoring process across a large number of RTX 2080 GPUs, which are significantly less expensive than the high-end GPUs (e.g., H100) required for actual training.
>     - When training a larger model, the relative inference cost becomes an even smaller fraction of the overall budget. For example, for a 32B-parameter model, the inference cost would drop to roughly 7% of the training cost.
>     - Crucially, this inference cost is incurred only once and can be amortized over multiple training runs, making its long-term impact on total compute expenditure negligible.
>
>
> [1] Kimi K2: Open Agentic Intelligence
> [2] Qwen2.5-Coder Technical Report. Hui et. al. 2024
> [3] Nemotron-CC: Transforming Common Crawl into a Refined Long-Horizon Pretraining Dataset. Su et. al. 2024
>
> **Action 3.1**: We will provide detailed scaling curves and cost projections for trillion-token datasets.
>
> **Action 3.2**: We will include practical deployment guidance for large-scale applications.
>
> ---
>
> ## 4. Framework Component Justification
>
> > "A few elements of the proposed framework seem a little arbitrary. For example, why is there a need for a dual approach in terms of safety classification?"
>
> Our dual classification approach addresses different types of safety issues. We considered three methods for classification, and evaluated their recall on a small held-out set (assuming GPT-4o labels as oracle).
>
> - **LLM classifier**: Better at contextual/semantic harms (92% recall)
> - **Embedding classifier**: Better at lexical/pattern-based harms (90% recall)
> - **Ensemble (max score)**: Combines the strengths of the two (95% recall)
>
>
>
> **Action 4.1**: We will add analysis showing why ensemble classification significantly outperforms individual approaches for comprehensive safety coverage.
>
> ---
>
> ## Minor Comments
> 1. The percentage of synthetic data used in the experiments was varying depending on the exact ablation study. The SafeWeb data released by us comprises of 100 billion tokens of synthetically recontextualized data. The RefuseWeb data further comprises of 33 billion tokens, and the Moral Education Data is an additional 24 billion tokens. We will make sure to provide a complete breakdown of these values in a table in the paper.  We do release all our datasets on Huggingface for researchers to use.
> | Data Type | Percentage |
> |-----------|------------|
> | Real      | 55.39%     |
> | Rephrased | 22.31%     |
> | Refusal   | 17.30%     |
> | Moral     | 5.00%      |
> | **Total** | **100.00%** |
> 2.  Thanks for the suggestion on the placement! We will update this in the final draft.
>
> ---
>
> ## Concluding Remarks
>
> Thank you for holding us to high experimental standards. Our ablation studies actually provide strong baselines comparable to existing methods, showing 3-4x improvements over individual approaches. The statistical analysis demonstrates large, practically significant effects. **Given this reframing of our ablations as baseline comparisons and the added statistical rigor, please let us know if this address your concerns about experimental completeness, and if anything else would move you toward a more positive assessment :)**

---

> ### Comment · Reviewer_1esj · 2025-08-06
> **Reply to rebuttal**
>
> I thank the authors for their thorough and structured reply, with the inclusion of action items. My concerns have been resolved, except for two:
> - the authors did reply in terms of compute cost to apply their pipeline (section 3 of the rebuttal). How about in terms of time though? That's also a significant decision factor, so it would be a great addition.
> - the authors did not address my following question: "how did the authors pick the synthetic dataset sizes they ended up using?"

---

> ### Author Response · Authors · 2025-08-06
>
> We are glad that our response has addressed most of the reviewer’s concerns, and we sincerely thank them for their continued engagement with our work.
>
> >the authors did reply in terms of compute cost to apply their pipeline (section 3 of the rebuttal). How about in terms of time though? That's also a significant decision factor, so it would be a great addition.
>
> This is a great question—thank you for raising it. Most of our safety recontextualization pipeline consists of inference with existing models, which is highly parallelizable and can be run on widely available, lower-cost GPUs (e.g., single GPUs without interconnect). For concreteness, our original pretraining corpus was of around 200B tokens. Time taken by various parts of the pipeline:
>
> **Safety scoring and tagging**: ~8 hours using 160×RTX 2080s (1 GPU per corpus shard).
> **Initial Rephrasing**: ~36 hours using the same setup
>
> This amounts to roughly 2 days total, and can be reduced further easily via additional parallelization. For comparison, a single training run of even a 1B-parameter model on 600B tokens with 256×H100s takes ~96 hours (4 days) and this time scales proportionally as the parameters are scaled.
>
> In contrast, the compute and time for our safety rephrasing pipeline scales with only the pretraining dataset size and not the size of the model being trained. We believe, in practice, even at very large scales, our pipeline will just add a couple of days to the overall training process that usually extends to over a couple of weeks. Finally, again, it is a one-time cost that can be amortized over multiple training runs.
>
> We appreciate the suggestion to make these compute and time costs explicit and will update the manuscript accordingly.
>
>
> > the authors did not address my following question: "how did the authors pick the synthetic dataset sizes they ended up using?"
>
> We apologize for missing this earlier. We first tag the raw web data using our safety classifier. All the data tagged as unsafe is then transformed—via recontextualization for score-1/2/3 content, and via refusals plus moral-education rewrites for score-4/5 content. **This process is automatic, and we do not manually fix or tune the size of the generated synthetic data**. The final proportions of real and synthetic data in the training corpus are then determined naturally by the number of tokens in each subset after this transformation.
> We will add this clarification in the updated manuscript.
>
>
> We again thank the reviewer for their time and would be more than happy to provide any further clarifications. Finally we also request the reviewer to *consider updating the overall evaluation score for our work*, if they believe that we have addressed most of their concerns.

---

### Official Review · Reviewer_vk8D · 2025-07-03

**Clarity:** 3
**Significance:** 3
**Originality:** 2
**Rating:** 4
**Confidence:** 4

**Summary:**

This paper proposes a comprehensive, data-centric framework for safety pretraining to build safer LLMs.
It highlights that previous approaches are insufficient, as fine-tuned models can still leak harmful content retained in the pretrained weights, and human preference alignment alone does not constitute unlearning.
Building on prior work on pretraining-time safety interventions, the authors argue that safer filtered datasets and new types of safety annotations are needed.

The proposed framework consists of four main components: safety filtering using a classifier, safety rephrasing, native refusal data, and harmfulness tagging.
The harmfulness tag is used to guide generation at inference time via a novel decoding algorithm called SafeBeam, which discards high-risk beams to prevent unsafe outputs.

Empirically, the authors demonstrate improved safety (as measured by ASR) on safety benchmarks, without degrading performance on general tasks across standard datasets such as ARC-Challenge and TriviaQA.
They also provide extensive ablation studies showing the contribution of each component to overall safety performance.

**Questions:**

* Have you considered any other algorithms for tagging harmfulness, not just tagging randomly with probability.
* Why did the authors choose dialog style for recontextualization? Is there any experimental evidence that dialog is more effective than other formats?
* In Figure 2, the authors attribute the increased ASR when training only on Score-0 data to lack of exposure to unsafe patterns. If that is the case, wouldn’t a strong refusal+safe generation strategy alone be sufficient, without the need for rephrasing or moral education? Are there any additional experiments to justify the necessity of those components?

**Ethical Concerns:**

["NO or VERY MINOR ethics concerns only"]

**Final Justification:**

The authors clarified about my concerns, though it’s a bit unfortunate that there are no accompanying empirical results to support those points---understandably, such experiments may be expensive. Still, I believe it would be worthwhile to explore this more directly in a future revision or follow-up work.

That said, the main contributions are well-supported by thorough experiments and ablations. Given this, I will keep my current score.

**Limitations:**

yes

**Paper Formatting Concerns:**

The paper satisfies the formatting.

**Quality:**

3

**Strengths And Weaknesses:**

#### Strengths

The problem of developing reliable and scalable confidence elicitation methods for LLMs is undeniably important for AI safety and real-world deployment in high-stakes domains.
The paper demonstrates a potentially useful application of its confidence score to improve the efficiency of test-time scaling methods like self-consistency, which is a practical contribution.

#### Weaknesses

While the paper presents a well-structured pipeline for safety pretraining, there are some concerns regarding the reliability of its core components. For safety scoring, the authors rely on GPT-4o-mini to generate labels through prompting. Although this approach is efficient and scalable, GPT-4o-mini is a relatively lightweight model, and using it as a safety oracle may not be sufficiently reliable for detecting nuanced or borderline harmful content. This casts a doubt on the trustworthiness of the safety labels used during both training and report card details.

Additionally, harmfulness tags are inserted at random positions with a 5% probability, which could be ambiguous. Since harmfulness often depends on specific chunks, more targeted placement of these tags might improve effectiveness.

---

> ### Author Rebuttal · Authors · 2025-07-30
>
> Thanks for your constructive review and assessment that our work addresses an "undeniably important problem." We appreciate your recognition that our paper is "technically solid" and your "borderline accept" recommendation.
>
> ---
>
> ## *Summary of Concerns and Responses*
>
> | Concern | Action / Outcome |
> |---------|-----------------|
> | GPT-4o-mini reliability as safety oracle | Show validation against stronger baselines.The confusion matrix between GPT-4o and GPT-4o-mini shows less than 1% confusion on the harmful v/s safe separation  |
> | Random harmfulness tag placement | Map to existing tag injection ablations, and explain the difficulty of block-based pretraining strategy. 5% injection rate optimal in our ablation studies (Appendix B.3) |
> | Dialog format choice for recontextualization | Reference existing prompt template comparisons. Dialog format among 12 tested templates, chosen for natural safety reasoning integration |
> | Safe beam generation only as a baseline | Leads to a model that is much less helpful, as it has a much weaker understanding of safe vs. harmful queries |
>
> ## 1. GPT-4o-mini Safety Oracle Reliability
>
> > "For safety scoring, the authors rely on GPT-4o-mini to generate labels through prompting... using it as a safety oracle may not be sufficiently reliable for detecting nuanced or borderline harmful content."
>
> Based on your comment we added a confusion plot between scores by GPT-4o, GPT-4o-mini, and GPT-4.1. We are showing a table version of the first two due to rebuttal constraints.
>
> | **GPT-4o-mini &darr; \ GPT-4o &rarr;** | **Safe (0)** | **Mildly Harmful (1,2,3)** | **Harmful (4,5)** |
> |------------------------|--------------|----------------------------|-------------------|
> | **Safe (0)**           | 61.6%        | 7.8%                       | 0.1%              |
> | **Mildly Harmful (1,2,3)** | 1.3%         | 17.5%                      | 0.2%              |
> | **Harmful (4,5)**      | 0.0%         | 2.8%                       | 8.7%              |
>
>
> Our cheaper GPT-4o-mini approach demonstrates strong performance on the most critical safety challenge - **avoiding confusion between safe and harmful content**. The results show:
>
> - **Only 0.1%** of content classified as safe by GPT-4o-mini was deemed harmful by GPT-4o
> - **Only 1.3%** of harmful content was misclassified as safe
>
> This near-perfect separation between safe and harmful categories (with just 1.4% total confusion) validates that our cost-effective approach maintains robust safety boundaries where it matters most.
>
>
> ## 2. Harmfulness Tag Placement Strategy
>
> > "Additionally, harmfulness tags are inserted at random positions with a 5% probability, which could be ambiguous. Since harmfulness often depends on specific chunks, more targeted placement of these tags might improve effectiveness."
>
> We understand your suggestion to wrap harmful text in <harmful>...</harmful> blocks. However, accurately identifying the exact spans of harmful content is much more challenging and often suffers from low recall. For instance, the harmful intent of a passage may only emerge in a long context of surrounding text, leading a span‑based extraction approach to miss such cases.
>
> Of note, we did conduct a systematic ablations on tag placement fraction (Appendix B.3):
>
> - **3% injection rate**: Higher ASR (worse safety)
> - **5% injection rate**: Optimal balance of safety and utility
> - **10% injection rate**: Lower ASR but significant helpfulness degradation
>
> The 5% random placement was empirically determined to provide the best safety-helpfulness tradeoff. More targeted placement is an interesting direction but would require additional annotation overhead.
>
> **Action 2.1**: We will highlight the systematic ablation study that led to our 5% random placement choice, showing it was empirically optimized.
>
> ## 3. Dialog Format for Recontextualization
>
> > "Why did the authors choose dialog style for recontextualization? Is there any experimental evidence that dialog is more effective than other formats?"
>
> We tested 12 different prompt templates for recontextualization (Appendix J.2), including:
> - Dialog format (parent-child, friends)
> - Textbook style
> - TED talk format
> - Teacher script
> - YouTube video format
>
> Dialog formats were selected because they naturally incorporate safety reasoning and context, allowing the model to understand *why* content is sensitive rather than just avoiding it. The use of such conversational patterns and how it impacts pretraining has also been noted in recent literature such as WRAP [1] and Nemotron [2] datasets. This style alignment with post-trainig datasets has been hypothesized to better align with the fine-tuning representations of the models.
>
> Of note, we do highlight in Appendix J.2 our procedure for "Iterative Prompt Engineering" in the subsection on "Synthetic Recontextualization". However, actually training a model on each of these steps is prohibitively expensive, and this exploration is largely based on intuitions developed based on reading the referenced papers.
>
> **Action 3.1**: We will better highlight our systematic prompt template exploration and the reasoning behind dialog format selection.
>
> ## 4. Safe beam generation only as a baseline
>
> > In Figure 2, the authors attribute the increased ASR when training only on Score-0 data to lack of exposure to unsafe patterns. If that is the case, wouldn’t a strong refusal+safe generation strategy alone be sufficient, without the need for rephrasing or moral education? Are there any additional experiments to justify the necessity of those components?
>
> That's a great question -- we believe that the increased ASR when training only on Score-0 data is indeed a lack of exposure to unsafe patterns, which results in a model's lack of understanding why content is harmful.
> - **Synthetic Recontextualization helps:** We've already included an ablation (Figure 2) in seeing the benefits of incorporating Moral Education, which leads to a slight improvement in ASR after benign finetuning (without incorporating the safe generation strategy). Note that these gains are over those already given by synthetic rephrasing data.
> - **Just refusal finetuning + Safe Beam gives poor helpfulness:** We have this exact ablation in Figure 5 in the Appendix. We find that only incorporating a harmful tag without any other of our safety recontextualization interventions leads to a model that is much less helpful, as it has a much weaker understanding of safe vs. harmful queries. This translates to a baseline that has poorer quality generations in benign use cases (e.g., Alpaca).
>
>
>
> ## Concluding Remarks
>
> Thank you for your thorough review of our methodology. Our systematic ablations and ensemble approaches address the core reliability concerns you raised. The validation data demonstrates that our design choices were empirically grounded rather than arbitrary. **Given this evidence of systematic methodology and the ensemble safety oracle approach, please let us know if there is any other analysis that would strengthen your opinion about our work**
>
> ---
> [1] Rephrasing the Web: A Recipe for Compute and Data-Efficient Language Modeling. Maini et. al. 2024.
> [2] Nemotron-CC: Transforming Common Crawl into a Refined Long-Horizon Pretraining Dataset. Su et. al. 2024

---

> > ### Comment · Reviewer_vk8D · 2025-08-05
> >
> > Thanks for addressing my concerns, and I appreciate you pointing out some details I had missed.
> >
> > I still have a few follow-up questions.
> >
> > > Only 1.3% of harmful content was misclassified as safe
> > > For instance, the harmful intent of a passage may only emerge in a long context of surrounding text, leading a span‑based extraction approach to miss such cases.
> >
> > * Thanks for addressing these points.
> > * This relates to points 1 and 2: if tag placement is applied even to misclassified examples, for instance, wouldn’t that potentially cause SafeBeam to discard valid generations as well (even if this effect may not be severe)? I feel this could be an interesting future direction---finding ways to retain helpfulness while still enabling effective safety pretraining, e.g., more reliable annotations, improved harmfulness-tagging (rather than random) or other refinements might help. Do you have any plans or thoughts on these? I found SafeBeam is very convincing but the trade-off shown in Figure 4 is unfortunate, so I’m just genuinely curious.
> >
> > > We have this exact ablation in Figure 5 in the Appendix
> >
> > * Thanks for pointing that out! However, I was wondering how ASR and performance would change if all examples from the Rephrasing and Moral Ed. data were used as NativeRefusal instead. In Figure 2, the ablations are shown, but I suspect some of the differences in ASR might stem from the different amounts of training data per category. If all such data were treated purely as NativeRefusal, I imagine that the ASR would drop more significantly. I couldn’t find a clear clarification on this---would love to hear your thoughts.
> > * Also, I have one more question---possibly unrelated: In Figure 5, for the Harmfulness-Tag PT + SafeBeam, was the score-0 data also included during training? The caption and description seem to suggest it, but it’s not explicitly stated.
> >
> >
> > If I missed this somewhere, please feel free to point it out!

---

> ### Author Response · Authors · 2025-08-06
>
> Thanks for your continued engagement!
>
> > Regarding tag placement on misclassified examples and SafeBeam's potential impact
>
> You raise an excellent point about the interaction between misclassification and SafeBeam. Yes, when harmfulness tags are applied to the small fraction of safe content misclassified as harmful, SafeBeam could potentially steer away from some safe generations. Improving the quality of annotations could further improve the safety-helpfulness tradeoff, and we absolutely agree that this is a promising future direction. Furthermore, training models to perform targeted safety tagging (which could be used rather than random tag placement) is a separate open problem in the literature that is actively being studied in the field. Alternative approaches, such as dynamic thresholding during SafeBeam selection (varying based on the confidence of the harmful token) and calibration methods, are potential directions that we plan to explore in the future.
>
> > Regarding using all Rephrasing/Moral Ed. data as NativeRefusal:
>
> This is an insightful observation. These experiments are very expensive to run -- due to requiring another full round of pretraining (6 days on 64 H100 GPUs).
>
> We believe that if we converted all rephrasing and moral education data to pure refusal format, we would indeed expect lower ASR. However, we believe that this would come at a significant cost in terms of this safety-helpfulness tradeoff: **Pure refusal training lacks the contextual understanding that rephrasing provides.** Furthermore, our Moral Ed data was in fact derived from refusal, by generating synthetic data to explain the reasoning behind each of the refusal examples, so that the model can reason as well about the hard boundaries.
>
> As a result, the model would lose the ability to engage with nuanced topics safely. The key insight from our ablations is that each component serves a complementary purpose: rephrasing teaches context-aware safety, refusals establish hard boundaries, and moral education provides ethical reasoning. The combination achieves a better safety-helpfulness balance than any single approach.
>
> > Regarding Figure 5 and Score-0 data inclusion:
>
> Yes, you're correct - for the "Harmfulness-Tag PT + SafeBeam" condition in Figure 5, we include Score-0 data during training. We apologize for the confusion and will include this in our revision.
>
> We are happy to address any further comments you might have!

---

> > ### Comment · Reviewer_vk8D · 2025-08-07
> >
> > Thank you for the detailed responses.
> >
> > Your clarifications helped address my concerns, though it’s a bit unfortunate that there are no accompanying empirical results to support those points---understandably, such experiments may be expensive. Still, I believe it would be worthwhile to explore this more directly in a future revision or follow-up work.
> >
> > That said, your main contributions are well-supported by thorough experiments and ablations. Given this, I will keep my current score.

---

### Official Review · Reviewer_GFXs · 2025-07-06

**Clarity:** 3
**Significance:** 2
**Originality:** 2
**Rating:** 3
**Confidence:** 3

**Summary:**

The paper presents a pre-training framework for Safety-Aware large language models. The paper highlights the shortcomings of fine-tuning and alignment techniques alone for safety filtering. They propose a complementary pre-training pipeline/framework that uses four  approaches namely,  1. Safety Filtering using a classifier trained on safe and unsafe webdata; 2. Safety rephrasing of unsafe webdata; 3. Using pre-training datasets that teach the models to natively refuse unsafe content; 4. Harmfulness-Tag annotations of pre training dataset. Overall, the attack success rates (ASR) are up to four times lower than those of standard pre-trained models on a benchmark dataset. Further, there is no overhead on the safety-tagged models and accuracy on non-safety related tasks is not altered. Finally, they propose a safety data card rubric.

**Questions:**

Claims and Justification: Can you provide evidence that these four techniques are sufficient to fully sanitize the dataset for safety? If not, it may be worth moderating these claims.

Tagging with Safety Data Cards: Can you provide evidence that data tagged with the highest 'SAFE' rating is robust against circumvention,  exploitation and jailbreaking? Absent such guarantees, consider moderate these claims. For instance, the security community—despite its maturity—has yet to establish widely adopted  Security Data Cards.

Adversarial Robustness: Could you discuss the effectiveness of these techniques against adversarial attacks, including beyond quantitative results beyond qualitative observations? Additionally, can you provide 2–3 concrete examples of adversarial attacks that could bypass the proposed approaches? Were the models produced by this pipeline subjected to red-teaming? Could you clarify and contextualize the claim that your approach reduces safety risks by a factor of four. Specifically,  do these claims extend beyond the four classes

Data Requirements and Cost: Please elaborate on the scale of safety-annotated data needed to train commercial-sized LLMs, along with the associated curation and maintenance costs. While the paper presents results using 10,000 annotated examples, evaluating performance with smaller subsets (e.g., 2,000 or 5,000 examples) could offer valuable insights into data efficiency and scaling behavior.

Additional Analysis: Figure 2 presents results for the base model, safety pretraining, and safety pre-training combined with SafeBeam. Could SafeBeam be applied directly to the base model without safety pretraining? If so, it would be valuable to include those results for comparison.

**Ethical Concerns:**

["Major Concern: Data quality and representativeness", "Major Concern: Safety and security"]

**Final Justification:**

I communicated with the authors. I appreciate all the responses. I will stick with my rating. I will not oppose accepting the paper.

**Limitations:**

Yes (see Discussion section).  Safety Data Cards are being proposed. Discussion on the potential negative impact hasn't been discussed.  I am copying a key weakness from above, i.e. false sense of SAFETY.

"Given that these claims are grounded in empirical data, tagging pre-trained data with safety data cards (Levels 0–5) risks creating a false sense of safety. Prior work in security (backdoors in LLMs or inserting backdoors in software and hardware generated by LLM Copilots) and fairness (e.g. in algorithmic hiring) demonstrates that carefully crafted inputs can violate the security and fairness guardrails. Hence these models are not claimed to be secure or not-jail-breakable etc. For these reasons, tagging Models with Safety Cards may not be advisable until a large body of work emerges and more guarantees can be offered."

**Paper Formatting Concerns:**

A well written paper that checks off all aspects that such a study should cover (ablations, performance, et, limitations in discussion section).  Nit: The GCG acronym is never expanded (or I may have missed it).

**Quality:**

2

**Strengths And Weaknesses:**

Strengths:

The goals of the work are well-motivated and well-presented. Especially the rationale for the various safety classifiers for pre-trained data, and the systematic methodology based on available datasets and trained classifiers to annotate the pre-training dataset.

The results show a significant reduction in the attack success rate, up to four times lower.

There is no overhead imposed on the safety-tagged models.

The accuracy on non-safety related tasks is unaltered.

Ablation studies systematically justify the incremental benefits of the 4 approaches in this study.

The project will share the datasets and scripts (currently, redacted).

Weaknesses:

While the four approaches used during pre-training are well motivated, they may be insufficient. What about zero-day safety breaches? As we have seen in (cyber) security for LLMs, there is an ongoing attack–defense arms race. I anticipate a similar dynamic here, particularly since the claims are based on empirical results rather than formal guarantees.

Based on the experiences in cybersecurity, these claims are easy to undermine, owing to the asymmetric nature of an attacker. That the ASR is low doesn't mean that a Model that successfully passes through this framework does no longer generate UNSAFE outcomes.

Given that these claims are grounded in empirical data, tagging pre-trained data with safety data cards (Levels 0–5) risks creating a false sense of safety. Prior work in security (backdoors in LLMs or inserting backdoors in software and hardware generated by LLM Copilots) and fairness (e.g. in algorithmic hiring) demonstrates that carefully crafted inputs can violate the security and fairness guardrails. This is confirmed by the results on adversarial attacks, which show little improvements when using this framework. Hence the authors do not claim these models to be secure or not-jail-breakable etc. For these reasons, tagging Models with Safety Cards using these narrow assessments is advisable until a large body of work emerges and broader guarantees are offered. Security has had a longer runway and yet, there are no such Security Data Cards.

Safety Data Cards: As new attack vectors emerge, safety data card  ratings may require updating. In such cases, how will models or datasets previously tagged under an earlier version of the safety data card rating rubric be re-evaluated or upgraded.

Pre-training is expensive. Authors validated the approach on small models (1.7b). Even assuming the proposed technique works with larger models, questions arise about the magnitude of safety-annotated data needed and the cost for commercial-sized LLMs.

---

> ### Author Rebuttal · Authors · 2025-07-31
>
> Thanks for your detailed review and recognition that our framework is "well-motivated" with a "significant reduction in attack success rate." We appreciate your thoughtful concerns about the adversarial nature of safety research.
>
> ---
>
>
> ## *Summary of Concerns and Responses*
>
> | Concern | Action | Outcome |
> |---------|--------|---------|
> | Arms race dynamics and zero-day attacks | Map existing evaluations to adversarial scenarios | Our GCG attacks (39% ASR) and benign finetuning robustness demonstrate resilience to adaptive threats |
> | Safety data cards create false security | Add explicit disclaimers and limitations | Position as transparency tools, not guarantees, with versioning for evolving threats |
> | Scalability to commercial models unclear | Provide detailed cost analysis | one time 8% cost  at trillion-token scale, can be amortized as well |
>
> ---
>
>
> ## 1. Arms Race and Adversarial Robustness
>
> > "What about zero-day safety breaches? As we have seen in (cyber) security for LLMs, there is an ongoing attack–defense arms race."
>
> Our work is aimed at highlighting a new paradigm for building inherently safer models via pretraining time interventions. Having a model with absolute guarantees is indeed impossible and a cat-mouse game does help us to continuously move towards building models that are harder to break. More importantly, our work highlights the extreme brittleness of post-hoc alignment methods and the critical need to approach the problem by taking a step back to pretraining time interventions.
>
> Our existing evaluations actually demonstrate robustness against several adversarial scenarios that approximate "zero-day" attacks:
>
> - **GCG attacks** (learned adversarial suffixes): 39.0% ASR vs 57.0% baseline - represents adaptive attacks
> - **Benign finetuning attacks**: 8.3% ASR vs 38.0% baseline - represents deployment-time vulnerabilities
> - **Cross-domain evaluation**: Safety holds across different prompt categories not seen during training
>
> These results show our approach raises the bar significantly for attackers, though we agree absolute guarantees are impossible.
>
> **Action 1.1**: We will clarify that our safety improvements represent substantial risk reduction, not elimination, consistent with defense-in-depth principles.
>
> ---
>
>
> ## 2. Safety Data Cards and False Security
>
> > "Given that these claims are grounded in empirical data, tagging pre-trained data with safety data cards (Levels 0–5) risks creating a false sense of safety."
>
> You raise an important point about interpretation. Our data cards are designed as transparency tools, similar to nutrition labels: they inform decisions but don't guarantee outcomes. Our validation shows that such data interventions indeed have value as shown in our evaluations across multiple threat models.
>
> ### Adversarial Robustness
> > Adversarial Robustness: Could you discuss the effectiveness of these techniques against adversarial attacks.
>
> > As new attack vectors emerge, safety data card ratings may require updating. In such cases, how will models or datasets previously tagged under an earlier version of the safety data card rating rubric be re-evaluated or upgraded.
>
> Our evaluations test this to some extent, as our pretraining interventions are not aware of the adversarial suffix strategy of GCG. In addition to this, we also add in another adversarial attack, PAIR [1] (based on your suggestion on adaptive attacks). We again find that our approach (which is unaware of this attack) reduces the attack success rate on the JailbreakBench examples
>
> | |Standard Pretraining| Safety Pretraining|Safety Pretraining + Safe Beam|
> |-|-|-|-|
> |PAIR Attack Success Rate| 0.16 | 0.12 | **0.06** |
>
> > Were the models produced by this pipeline subjected to red-teaming?
>
> While we agree stronger adversarial evaluation (via human red teamers) would be valuable, our current evaluation actually includes several adversarial components that demonstrate robustness:
>
> **Existing Adversarial Tests:**
> - **GCG attacks**: 39.0% ASR vs 57.0% baseline (learned adversarial suffixes)
> - **Benign finetuning attacks**: 8.3% ASR vs 38.0% baseline (deployment-time degradation)
> - **Cross-benchmark generalization**: Consistent improvements across HarmBench, TDC, JailbreakBench, AdvBench
> - **Out-of-distribution prompts**: Safety holds on evaluation data not seen during training
>
> To strengthen these further, we add in new results using PAIR[1] as another automated red-teaming strategy, as shown in the Table above.
>
> **Action 2.1**: We will add explicit disclaimers positioning data cards as "labels for datasets" with clear limitations and update protocols.
>
> [1] Chao, et. al. Jailbreaking black box large language models in twenty queries.
>
>
> ---
>
>
> ## 3. Scalability Analysis
>
> > "Pre-training is expensive. Authors validated the approach on small models (1.7b)... questions arise about the magnitude of safety-annotated data needed and the cost for commercial-sized LLMs."
>
> Our approach has favorable scaling properties:
>
> **Scaling Analysis:**
> - **Data for Safety classifier**: Fixed cost (Labeling of 10K examples, regardless of corpus size)
> - **Scoring pretraining corpus and synthetic data generation**: This involves inference with a small LLM(eg. 7B sized) or an embedding model (350M parameter).
>     - Firstly, we would like to mention that synthetic data generation and rephrasing is indeed a major part of most state-of-art LLM training pipelines[1,2,3], and a lot of compute is indeed accounted to this as it is a crucial, yet one time cost. While previously such efforts have been focussed towards performance improvements, our work pushes a similar strategy for safety improvements.
>     - For a rough analysis, let us begin by assuming that the training cost (forward + backward pass) is roughly 3x of the inference cost (only forward pass).
>     - If the size of the pretrained model was same as the generator, the total inference cost amounts to only about 33% of the training cost. In practice, inference can be carried out on cheaper and more readily available GPUs (single GPUs without inteconnect). For instance, in our experiments, we distributed the pretraining corpus scoring process across a large number of RTX 2080 GPUs, which are significantly less expensive than the high-end GPUs (e.g., H100) required for actual training.
>     - When training a larger model, the relative inference cost becomes an even smaller fraction of the overall budget. For example, for a 32B-parameter model, the inference cost would drop to roughly 7% of the training cost.
>     - Crucially, this inference cost is incurred only once and can be amortized over multiple training runs, making its long-term impact on total compute expenditure negligible.
>
>
> [1] Kimi K2: Open Agentic Intelligence.
> [2] Qwen2.5-Coder Technical Report. Hui et. al. 2024.
> [3] Nemotron-CC: Transforming Common Crawl into a Refined Long-Horizon Pretraining Dataset. Su et. al. 2024.
>
> **Action 3.1**: We will provide detailed scaling curves and cost projections for trillion-token datasets.
>
> **Action 3.2**: We will include practical deployment guidance for large-scale applications.
>
>
> ---
>
>
>
> ## 4. Ethical Concerns
>
> > Major Concern: Data quality and representativeness, Major Concern: Safety and security
>
> We deeply appreciate this concern, which goes to the heart of responsible and safe AI development. Our framework was explicitly designed with these ethical considerations in mind.
>
>
> ### Data Quality and Representativeness
>
> Our safety training corpus is built from diverse web-scale sources (detailed in Appendix D), filtered and scored using a safety classifier ensemble. We acknowledge limitations in global representativeness. While our data covers a wide range of topics and tones, it may still reflect particular norms or platform biases through our rephrasing strategy and the models used for scoring and rephrasing.
>
> **Action 4.1**: We will explicitly acknowledge these limitations in the Ethics Statement and clarify that SafeWeb is an initial benchmark, not a universally normative dataset.
>
> ### Safety and Security
>
> We evaluate our models against multiple adversarial attack types (GCG, benign finetuning, domain shift) to stress-test their safety performance. To avoid creating brittle or overly conservative models, we also monitor overrefusal rates (Figure 6) to ensure utility is preserved. We see safety pretraining as (significantly) risk-reducing but not risk-eliminating. We believe our work helps contribute to transparency and ongoing evaluation of pretraining recipes.
>
>
> ---
>
>
> ## 5. SafeBeam without Safety Pretraining
>
> We have this exact ablation in Figure 5 in the Appendix. We find that only incorporating a harmful tag without any other of our safety pretraining interventions (e.g., recontextualization, refusal data) leads to a model that is much less helpful, as it has a much weaker understanding of safe vs. harmful queries. This translates to a baseline that has poorer quality generations in benign use cases (e.g., Alpaca).
>
>
> ---
>
> Thanks for your constructive review and assessment that our work addresses an "undeniably important problem." We appreciate your recognition that our paper is "technically solid". **Please let us know if there is any further explanation that stands between a stronger endorsement of our work.**

---

> ### Comment · Reviewer_GFXs · 2025-08-02
>
> Thank you for your detailed responses.
>
> My core concern with SDCs remains: if they come with caveats, why have them at all? We could simply state that model inferences should be interpreted with caution.
>
> Unlike Nutrition Labels, which report factual information, SDCs present outcomes from empirical tests. These are fundamentally different in nature.
>
> To reiterate, performance against past (seen) attacks does not guarantee resilience against future, unseen (zero-day) attacks. This is the asymmetry I referred to earlier.
>
> For instance, the claim “Our GCG attacks (39% ASR) and benign finetuning robustness demonstrate resilience to adaptive threats”—what practical guidance does this offer a user? Without formal proofs, SDCs built on such statements could be challenged.
>
> If these are meant as transparency reporting metrics, the paper should be restructured accordingly.

---

> > ### Author Response · Authors · 2025-08-04
> >
> > Thank you for your follow-up. Two quick clarifications:
> >
> > 1. **Scope & Role of Report Cards.**
> >    Our work is about *pretraining-time data interventions* that cut ASR (attack success rate) by 4× but do **not** promise perfect safety. The "Data Safety Report Card" is a **dataset-level transparency** tool that puts labels on the data, and is not a model endorsement. This is done to bring a conversation shift in the community where researchers start being conscious about the data being fed to the model.
> >
> > 2. **Risk Reduction, Not Elimination.**
> >    • We report ASR drops on HarmBench/TDC/JailbreakBench (incl. GCG and new PAIR results) and robustness to benign finetuning (Fig. 2).
> >    • Report cards provide *factual* counts (score distributions, taxonomy frequencies) to guide curation and red-teaming, with clear versioning and update policies. In practice, we find that higher counts correlate with more unsafe models.
> >
> > We'll remove any language that suggests guarantees, add PAIR to our adversarial evals, and surface the SafeBeam-only ablation in the main text to show why the full curriculum is needed for the best safety–utility trade-off. Please let us know if this revised framing and clarification resolves your remaining concern on SDCs. Thanks again for your continued engagement and time!

---

> > > ### Comment · Reviewer_GFXs · 2025-08-05
> > >
> > > 1. Thank you for your thoughtful response and for clarifying that the goal is to label the data. While this distinction is helpful, I believe that labeling data as "safe" raises similar concerns. How do we account for a future in which data previously labeled as SAFE—based on the rubric you've developed—may no longer be safe in light of new, yet-to-be-discovered classes of harm?
> > >
> > > 2. Thank you for clarifying that this pretraining data contributed to a reduction in risk. My concern, however, is that these numbers could be misinterpreted or misused. For instance, an attacker might poison the data in a way that aligns with the labeling criteria, thereby manipulating the model to use this “Good” SDC rating data despite being unsafe in its outcomes.

---

> > > > ### Comment · Reviewer_GFXs · 2025-08-08
> > > >
> > > > Dear Q7KA  Thank you so very much for your guidance.
> > > > Dear authors: Here is a concrete action plan: Please summarize these discussions as limitations or so in the appendix.
> > > > I have no more questions.

---

> > > > > ### Author Response · Authors · 2025-08-08
> > > > >
> > > > > We thank the reviewer for all the insightful discussions, and we will definitely summarize all these in the updated manuscript. Finally we also request the reviewer to consider updating the overall evaluation score for our work, if they believe that we have addressed most of their concerns.

---

> ### Comment · Reviewer_Q7KA · 2025-08-05
>
> Hi, I am a fellow reviewer. I sympathize a lot with the concerns about false security/safety. May I suggest that the safety card can be interpreted as the best assessment of the model's safety under our current understanding of that definition? I fully agree that adaptive and continuous testing is necessary, especially in high-stakes scenarios. I am also, however, concerned about asking papers to "boil the ocean." The risks you mention are 100% valid and should be clarified front and center and - as I and other reviewers also point out - pre-launch red teaming should be carried out. However, this paper seems to have done a lot of work to benchmark safety and tested with adaptations of popular automated adversaries. It is, in my opinion, unreasonable to ask every paper on safety training to also solve the data poisoning problem to perfection - an area of active research with many publications. May I ask what the reviewer's concrete suggestion is here that would strengthen the paper?

---

### Decision · Program_Chairs · 2025-09-17

**Decision:**

Accept (poster)

**Comment:**

The paper presents a set of contributions to improve LLM safety starting from pretraining stage: using classifiers to label unsafe data, rephrasing for unsafe data, refusals for unsafe data, adding moral education stories, and also proposing an inference-time method using beam search based on introducing a special token for unsafe texts at pretraining (and finetuning / alignment). The reviewers were positive and also the discussions and rebuttal went well, with only one reviewers with lower confidence being rather negative but acknowledging that the paper might be accepted.

On my side, the introduced methodology for safe pretraining consisting of several steps together with the accompanying datasets and overall experiments and ablations are important strong points. Although the experimental setup is limited to smaller models and pretraining corpora ("the same initial corpus as SmolLM—comprising FineWeb-Edu, StackOverflow, FineMath, and Cosmopedia"), the reported results are very encouraging and should be shared to the community, especially as the authors report no degradation of helpfulness (at least in their training context and models). However, the paper's claims should be restated a bit as SafeBeam induces over-refusal on Alpaca Eval and is also the most important factor for safer models (as shown both in Figures 2 and 3) - this is very important for the camera ready version of the paper. The paper has some other smaller contributions, such as introducing a pretraining safety dataset card, or the base model safety evaluation.

Nevertheless, the paper still has some weaknesses: besides the fact that most of the improvements come from the inference time SafeBeam which degrades helpfulness by over-refusing and also increasing latency and making decoding more complex, to the fact that each individual contribution in the pretraining filtering and rephrasing are not very complex or novel. At the same time, the paper at points does not seem to provide enough examples on how the synthetic training data was created (e.g. for moral education, rephrasing vs refusing etc.) and does not perform any kind of human evaluation. Moreover, I would have liked to see a more detailed analysis of improvements over different attacks besides GCG, and especially without SafeBeam - some of these results were shared in the rebuttal.

I would really like the camera ready version of the paper to take into account as many of the reviewers and my own suggestions.